# Make the U in UDA Matter: Invariant Consistency Learning for Unsupervised Domain Adaptation

**Zhongqi Yue[1],    Hanwang Zhang[1],    Qianru Sun[2]**
[1]Nanyang Technological University,    [2]Singapore Management University
yuez0003@ntu.edu.sg,    hanwangzhang@ntu.edu.sg,    qianrusun@smu.edu.sg

## Abstract

Domain Adaptation (DA) is always challenged by the spurious correlation between domain-invariant features (*e.g.*, class identity) and domain-specific features (*e.g.*, environment) that does not generalize to the target domain. Unfortunately, even enriched with additional unsupervised target domains, existing Unsupervised DA (UDA) methods still suffer from it. This is because the source domain supervision only considers the target domain samples as auxiliary data (*e.g.*, by pseudo-labeling), yet the inherent distribution in the target domain—where the valuable de-correlation clues hide—is disregarded. We propose to make the U in UDA matter by giving equal status to the two domains. Specifically, we learn an invariant classifier whose prediction is simultaneously consistent with the labels in the source domain and clusters in the target domain, hence the spurious correlation inconsistent in the target domain is removed. We dub our approach "Invariant CONsistency learning" (ICON). Extensive experiments show that ICON achieves the state-of-the-art performance on the classic UDA benchmarks: OFFICE-HOME and VISDA-2017, and outperforms all the conventional methods on the challenging WILDS 2.0 benchmark. Codes are in Appendix.

## 1   Introduction

Domain Adaptation (DA) is all about training a model in a labelled source domain $S$ (*e.g.*, an autopilot trained in daytime), and the model is expected to generalize in a target domain $T$ (*e.g.*, the autopilot deployed in nighttime), where $T$ and $S$ has a significant domain shift (*e.g.*, day *vs.* night) [41]. To illustrate the shift and generalization, we introduce the classic notion of causal representation learning [20, 49]: any sample is generated by $\mathbf{x} = \Phi(\mathbf{c}, \mathbf{e})$, where $\Phi$ is the generator, $\mathbf{c}$ is the causal feature determining the domain-invariant class identity (*e.g.*, road lane), and $\mathbf{e}$ is the environmental feature (*e.g.*, lighting conditions). As the environment is domain-specific, the domain shift $\mathbf{e}_S \neq \mathbf{e}_T$ results in $\mathbf{x}_S \neq \mathbf{x}_T$, challenging the pursuit of a domain-invariant

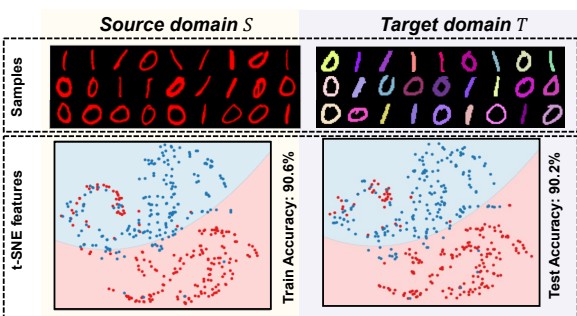

Figure 1: A digit classification model trained on red digits in $S$ generalizes to colorful digits in $T$ by disentangling digit shape (*i.e.*, the causal feature $\mathbf{c}$), as color in $S$ (*i.e.*, $\mathbf{e}_S$) is not discriminative. ●: "0", ●: "1". The classification boundary is rendered blue and red.

class model. However, if a model disentangles the causal feature by transforming each $\mathbf{x} = \mathbf{c}$, it can generalize under arbitrary shift in $\mathbf{e}$. In Figure 1, even though $\mathbf{e}_T$ = "colorful" is significantly

37th Conference on Neural Information Processing Systems (NeurIPS 2023).

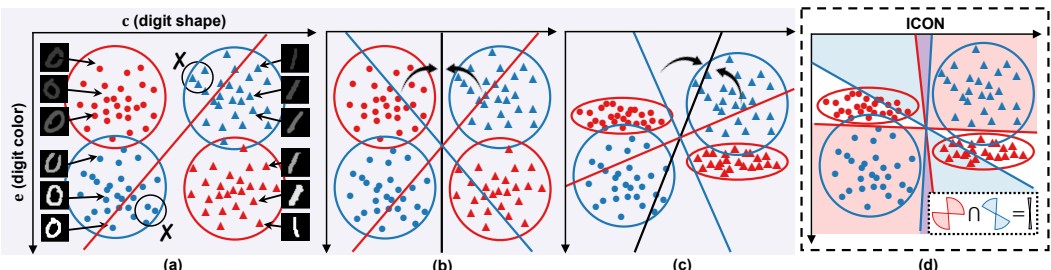

Figure 2: Red: $S$, Blue: $T$. ●: "0", ▲: "1". (a) A failure case of existing lineups. The red line denotes the classifier trained in $S$. (b) Classifier can be corrected (black line) by respecting the BCE loss between $S$ (red line) and $T$ (blue line). (c) A failure case of minimizing the combined BCE losses in $S$ and $T$. Due to the low variance of $\mathbf{e}_S$ in each class, the optimal classifier *w.r.t.* the BCE loss in $S$ (red line) looks "flat", *i.e.*, it has large weight on $\mathbf{e}$ [39]. (d) Our ICON prevents such a failure case.

different from $\mathbf{e}_S =$ "red", thanks to the non-discriminative "red" color in $S$, the model still easily disentangles $\mathbf{c}$ from $\Phi(\mathbf{c}, \mathbf{e}_S)$ and thus generalizes to digits of any color.

Unfortunately, in general, disentangling $\mathbf{c}$ by using only $S$ is impossible as $\mathbf{e}_S$ can be also discriminative, *i.e.*, spuriously correlated with $\mathbf{c}$. In Figure 2a, the color $\mathbf{e}_S$ is correlated with the shape $\mathbf{c}$, *i.e.*, $\mathbf{c}_1 =$"0" and $\mathbf{c}_2 =$"1" in $S$ tend to have $\mathbf{e}_1 =$ dark and $\mathbf{e}_2 =$ light digit colors, respectively. As both $\mathbf{c}$ and $\mathbf{e}_S$ are discriminative, a classifier trained in $S$ will inevitably capture both (see the red line in Figure 2a). This leads to poor generalization when the correlation between $\mathbf{e}_T$ and $\mathbf{c}$ is different, *i.e.*, the colors of $\mathbf{c}_1 =$"0" and $\mathbf{c}_2 =$"1" in $T$ tend to be $\mathbf{e}_2 =$ light and $\mathbf{e}_1 =$ dark instead.

To avoid spurious correlations between $\mathbf{e}_S$ and $\mathbf{c}$, a practical setting called Unsupervised DA (UDA) introduces additional information about $T$ through a set of unlabeled samples [15, 35, 12]. There are two main lineups: 1) Domain-alignment methods [58, 34, 10] learn the common feature space of $S$ and $T$ that minimizes the classification error in $S$. In Figure 2a, their goal is to unify "0"s and "1"s in $S$ and $T$. However, they suffer from the well-known misalignment issue [31, 71], *e.g.*, collapsing "0"s and "1"s in $T$ to "1"s and "0"s in $S$ respectively also satisfies the training objective, but it generalizes poorly to $T$. 2) Self-training methods [72, 38, 52] use a classifier trained in $S$ to pseudo-label samples in $T$, and jointly train the model with the labels in $S$ and confident pseudo-labels in $T$. Yet, the pseudo-labels can be unreliable even with expensive threshold tweaking [32, 29], *e.g.*, the ✗ areas in Figure 2a have confident but wrong pseudo-labels. In fact, the recent WILDS 2.0 benchmark [47] on real-world UDA tasks shows that both lineups even under-perform an embarrassingly naive baseline: directly train in $S$ and test in $T$.

Notably, although both methods respect the idiosyncrasies in $S$ (by minimizing the classification error), they fail to account for the inherent distribution in $T$, *e.g.*, in Figure 2a, even though $T$ is unlabelled, we can still identify the two sample clusters in $T$ (enclosed by the blue circles), which provide additional supervision for the classifier trained in $S$. In particular, the classifier (red line) in Figure 2a breaks up the two clusters in $T$, showing that the correlation between color $\mathbf{e}_S$ and $\mathbf{c}$ in $S$ is inconsistent with the clusters affected by $\mathbf{e}_T$ in $T$, which implies that color $\mathbf{e}$ is the environmental feature. Hence, to make the U in UDA matter, we aim to learn a classifier that is consistent with classification in $S$ and clustering in $T$:

**Consistency**. We use the Binary Cross-Entropy (BCE) loss to encourage the prediction similarity of each sample pair from the same class of $S$ or cluster of in $T$, and penalize the pairs from different ones. In Figure 2b, the red and blue lines denote the classifiers that minimize the BCE loss in $S$ and $T$, respectively. By minimizing the combined loss, we learn the classifier consistent with the sample distribution in both domains (the upright black line), which predicts solely based on the causal feature $\mathbf{c}$ (*i.e.*, disentangling $\mathbf{c}$ from $\Phi(\mathbf{c}, \mathbf{e})$).

**Invariance**. However, enforcing consistency is a tug-of-war between $S$ and $T$, and the BCE loss in one domain can dominate the other to cause failures. Without loss of generality, we consider the case where $S$ dominates in Figure 2c. Due to the strong correlation between $\mathbf{e}_S$ and $\mathbf{c}$, the classifier that minimizes the combined losses of BCE (black line) deviates from the desired upright position as in Figure 2b. To tackle this, we aim to learn an invariant classifier [1] that is simultaneously optimal *w.r.t.* the BCE loss in $S$ and $T$. In Figure 2d, for each of the two losses, we color the candidate regions where the classifier is optimal (*i.e.*, gradients $\approx 0$) with red and blue, respectively. In particular, by

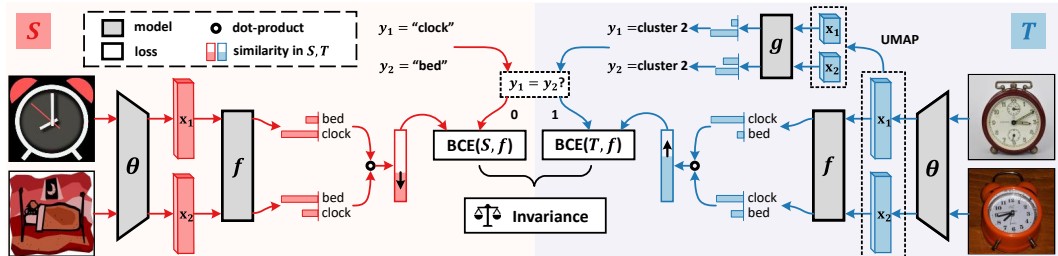

Figure 3: A training example where the pair of samples in $S$ is from different classes and that in $T$ is from the same cluster. The black arrows on the similarity bars denotes that minimizing the BCE losses in $S$ and $T$ will decrease the similarity in $S$ (as $y_1 \neq y_2$) and increase that in $T$ (as $y_1 = y_2$).

learning an invariant classifier lying in the intersection (purple region), we successfully recover the desired one as in Figure 2b, even when one domain dominates.

Overall, we term our approach as **Invariant CONsistency learning (ICON)**. Our contributions:

- ICON is a novel UDA method that can be plugged into different self-training baselines. It is simple to implement (Section 2), and disentangles the causal feature $\mathbf{c}$ from $\Phi(\mathbf{c}, \mathbf{e})$ that generalizes to $T$ with a theoretical guarantee (Section 3).
- ICON significantly improves the current state-of-the-art on classic UDA benchmarks [29, 6] with ResNet-50 [19] as the feature backbone: 75.8% (+2.6%) accuracy averaging over the 12 tasks in OFFICE-HOME [60], and 87.4% (+3.7%) mean-class accuracy in VISDA-2017 [43] (Section 5).
- Notably, as of NeurIPS 2023, ICON dominates the leaderboard of the challenging WILDS 2.0 UDA benchmark [47] (https://wilds.stanford.edu/leaderboard/), which includes 8 large-scale classification, regression and detection tasks on image, text and graph data (Section 5).

## 2 Algorithm

UDA aims to learn a model with the labelled training data $\{\mathbf{x}_i, y_i\}_{i=1}^N$ in the source domain $S$ and unlabelled $\{\mathbf{x}_i\}_{i=1}^M$ in the target domain $T$, where $\mathbf{x}_i$ denotes the feature of the $i$-th sample extracted by a backbone parameterized by $\theta$ (*e.g.*, ResNet-50 [19]), and $y_i$ is its ground-truth label. We drop the subscript $i$ for simplicity when the context is clear. As shown in Figure 3, our model includes the backbone $\theta$, a classification head $f$, and a cluster head $g$, where $f$ and $g$ output the softmax-normalized probability for each class and cluster, respectively. Note that $f$ and $g$ have the same output dimension as the class numbers are the same in $S$ and $T$. As discussed in Section 1, the generalization objective of ICON is to learn an invariant $f$ that is simultaneously consistent with the classification in $S$, and the clustering in $T$, identified by $g$. ICON is illustrated in Figure 3 and explained in details below.

### 2.1 Consistency with $S$

We use Binary Cross-Entropy (BCE) loss to enforce the consistency of $f$ with $S$, denoted as $\mathrm{BCE}(S, f)$. Specifically, $\mathrm{BCE}(S, f)$ is a pair-wise loss given by:

$$\mathrm{BCE}(S, f) = \mathbb{E}_{(\mathbf{x}_i, \mathbf{x}_j) \sim S} \left[ b \log(\hat{y}) + (1 - b) \log(1 - \hat{y}) \right], \tag{1}$$

where $b = \mathbb{1}(y_i = y_j)$ is a binary label indicating if the pair $(\mathbf{x}_i, \mathbf{x}_j)$ is from the same class, and $\hat{y} = f(\mathbf{x}_i)^\mathsf{T} f(\mathbf{x}_j)$ is the predictive similarity. Intuitively, BCE is a form of contrastive loss that clusters sample pairs with the same label (by increasing the dot-product similarity of their predictions) and pushes away pairs with different labels (by decreasing the similarity). Note that in practice, BCE loss is computed within each mini-batch, hence there is no $N^2$ prohibitive calculation.

### 2.2 Consistency with $T$

Unfortunately, $\mathrm{BCE}(T, f)$ cannot be evaluated directly, as the binary label $b$ in Eq. (1) is intractable given unlabeled $T$. To tackle this, we cluster $T$ to capture its inherent distribution and use cluster labels to compute $\mathrm{BCE}(T, f)$.

**Clustering** $T$. We adopt the rank-statistics algorithm [18], as it can be easily implemented in a mini-batch sampling manner for online clustering. The algorithm trains the cluster head $g$ to group features whose top values have the same indices (details in Appendix). Ablation on clustering algorithm is in Section 5.4. Note that our $g$ is trained from scratch, rather than initialized from $f$ trained in $S$ [56, 29, 32]. Hence $g$ captures the distribution of $T$ without being affected by that of $S$.

**Computing** $\text{BCE}(T, f)$. After training $g$, we replace $b = \mathbb{1}(y_i = y_j)$ for $\text{BCE}(S, f)$ in Eq. (1) with $b = \mathbb{1}(\arg\max g(\mathbf{x}_i) = \arg\max g(\mathbf{x}_j))$ for $\text{BCE}(T, f)$, which compares the cluster labels of each sample pair. We emphasize that it is necessary to use pairwise BCE loss to enforce the consistency of $f$ with $T$. This is because cross-entropy loss is not applicable when the cluster labels in $T$ are not aligned with the class indices in $S$.

## 2.3 Invariant Consistency (ICON)

ICON simultaneously enforces the consistency of $f$ with $S$ (*i.e.*, the decision boundary of $f$ separates the classes) and $T$ (*i.e.*, it also separates the clusters). The objective is given by:

$$\min_{\theta, f} \overbrace{\text{BCE}(S, f) + \text{BCE}(T, f)}^{\text{Consistency}} + \overbrace{\text{CE}(S, f) + \alpha\mathcal{L}_{st}}^{S\text{-Supervision}}$$
$$\text{s.t.} \quad \underbrace{f \in \arg\min_{\bar{f}}\text{BCE}(S, \bar{f}) \cap \arg\min_{\bar{f}}\text{BCE}(T, \bar{f})}_{\text{Invariance}}. \tag{2}$$

**Line 1**. The BCE losses train $f$ to be consistent with the pair-wise label similarity in $S$ and $T$. CE loss in $S$ trains $f$ to assign each sample its label (*e.g.*, predicting a clock feature as "clock"). $\mathcal{L}_{st}$ is the self-training loss with weight $\alpha$, which leverages the learned invariant classifier to generate accurate pseudo-labels in $T$ for additional supervision (details in Section 5.2).

**Line 2**. The constraint below requires the backbone $\theta$ to elicit an invariant classifier $f$ that simultaneously minimizes the BCE loss in $S$ and $T$. This constraint prevents the case where one BCE loss is minimized at the cost of increasing the other to pursue a lower sum of the two (Figure 2c).

To avoid the challenging bi-level optimization, we use the following practical implementation:

$$\min_{\theta, f} \quad \text{BCE}(S, f) + \text{BCE}(T, f) + \text{CE}(S, f) + \alpha\mathcal{L}_{st} + \beta\text{Var}\left(\{\text{BCE}(S, f), \text{BCE}(T, f)\}\right), \quad (3)$$

where the variance term $\text{Var}(\cdot)$ is known as the REx loss [25] implementing the invariance constraint. The self-training weight $\alpha$ and invariance weight $\beta$ are later studied in ablation (Section 5.4).

**Training and Testing**. Overall, at the start of training, the backbone $\theta$ is initialized from a pre-trained weight (*e.g.*, on ImageNet [46]) and $f, g$ are randomly initialized. Then ICON is trained by Eq. (3) until convergence. Only the backbone $\theta$ and the classifier $f$ are used in testing.

## 3 Theory

**Preliminaries**. We adopt the group-theoretic definition of disentanglement [20, 62] to justify ICON. We start by introducing some basic concepts in group theory. $\mathcal{G}$ is a group of semantics that generate data by group action, *e.g.*, a "turn darker" element $g \in \mathcal{G}$ transforms $\mathbf{x}$ from white "0" to $g \circ \mathbf{x}$ as grey "0" (bottom-left to top-left in Figure 4b). A sketch of theoretical analysis is given below and interested readers are encouraged to read the full proof in Appendix.

**Definition** (Generalization). *Feature space $\mathcal{X}$ generalizes under the direct product decomposition $\mathcal{G} = \mathcal{G}/\mathcal{H} \times \mathcal{H}$, if $\mathcal{X}$ has a non-trivial $\mathcal{G}/\mathcal{H}$-representation, i.e., the action of each $h \in \mathcal{H}$ corresponds to a trivial linear map in $\mathcal{X}$ (i.e., identity map), and the action of each $g \in \mathcal{G}/\mathcal{H}$ corresponds to a non-trivial one.*

This definition formalizes the notion of transforming each sample $\Phi(\mathbf{c}, \mathbf{e})$ to the causal feature $\mathbf{c}$. Subgroup $\mathcal{H} \subset \mathcal{G}$ and quotient group $\mathcal{G}/\mathcal{H}$ transform the environmental feature $\mathbf{e}$ and the causal $\mathbf{c}$, respectively. If $\mathcal{X}$ has a non-trivial $\mathcal{G}/\mathcal{H}$-representation, samples in a class are transformed to the same feature $\mathbf{c} \in \mathcal{X}$ regardless of $\mathbf{e}$ (*i.e.*, representation of $\mathcal{G}/\mathcal{H}$), and different classes have different $\mathbf{c}$ (*i.e.*, non-trivial). This definition is indeed in accordance with the common view of good features [62], *i.e.*, they enable zero-shot generalization as $\mathbf{c}$ is class-unique, and they are robust to the

domain shift by discarding the environmental features $\mathbf{e}$ from $\Phi(\mathbf{c}, \mathbf{e})$. To achieve generalization, we need the following assumptions in ICON.

**Assumption** (Identifiability of $\mathcal{G}/\mathcal{H}$).
*1)* $\forall \mathbf{x}_i, \mathbf{x}_j \in T$, $y_i = y_j$ *iff* $\mathbf{x}_i \in \{h \circ \mathbf{x}_j \mid h \in \mathcal{H}\}$;
*2) There exists no linear map* $l : \mathcal{X} \to \mathbb{R}$ *such that* $l(\mathbf{x}) > l(h \circ \mathbf{x})$, $\forall \mathbf{x} \in S, h \circ \mathbf{x} \in T$ *and* $l(g \circ \mathbf{x}) < l(gh \circ \mathbf{x})$, $\forall g \circ \mathbf{x} \in S, gh \circ \mathbf{x} \in T$, *where* $h \neq e \in \mathcal{H}, g \neq e \in \mathcal{G}/\mathcal{H}$.

Assumption 1 states that classes in $T$ are separated by clusters. This corresponds to the classic clustering assumption, a necessary assumption for learning with unlabeled data [59]. Figure 4a shows a failure case of Assumption 1, where the intra-class distance (*i.e.*, dotted double-arrow) is much larger than the inter-class one (*i.e.*, solid double-arrow) in $T$, *i.e.*, shape $\mathbf{c}$ is less discriminative than color $\mathbf{e}_T$. Hence the clusters in $T$ are based on $\mathbf{e}$, causing ICON to capture the wrong invariance. To help fulfill the assumption, UDA methods (and SSL in general) commonly leverage feature pre-training and data augmentations. We also specify the number of clusters as the class numbers in $T$, which is a necessary condition of this assumption.

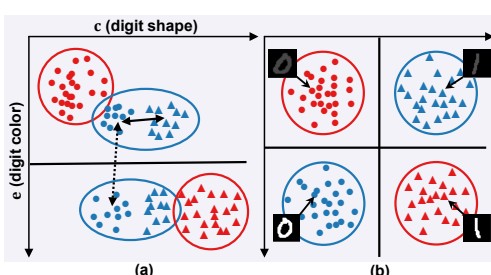

Figure 4: Red: $S$, Blue: $T$. ●: "0", ▲: "1". Black lines denote the invariant classifier satisfying the ICON objective. (a) Assumption 1 is violated. (b) Assumption 2 is violated.

Assumption 2 states that $\mathbf{c}$ is the only invariance between $S$ and $T$. Figure 4b illustrates a failure case, where there exist two invariant classifiers: both the vertical one based on shape $\mathbf{c}$ and the horizontal one based on color $\mathbf{e}$ separate the classes in $S$ and clusters in $T$. Yet as $T$ is unlabelled, there is no way for the model to determine which one is based on $\mathbf{c}$. In practice, this assumption can be met by collecting more diverse unlabelled samples in $T$ (*e.g.*, darker "0"s and lighter "1"s in Figure 4b), or introducing additional priors on what features should correspond to the causal one.

**Theorem**. *When the above assumptions hold, ICON (optimizing Eq. (3)) learns a backbone $\theta$ mapping to a feature space $\mathcal{X}$ that generalizes under $\mathcal{G}/\mathcal{H} \times \mathcal{H}$. In particular, the learned $f$ is the optimal classifier in $S$ and $T$.*

By transforming each sample $\Phi(\mathbf{c}, \mathbf{e})$ in $S$ and $T$ to $\mathbf{c} \in \mathcal{X}$, the classifier easily associate each $\mathbf{c}$ to its corresponding class label using the labelled $S$ to reach optimal.

# 4    Related Work

**Domain-Alignment** methods [13, 4, 57] map samples in $S$ and $T$ to a common feature space where they become indistinguishable. Existing works either minimize the discrepancy between the sample distributions of $S$ and $T$ [36, 2, 27], or learn features that deceive a domain discriminator [69, 63, 33]. However, they suffers from the misalignment issue under large domain shift from $S$ to $T$ [71, 31], such as shift in the support of the sample distribution [23] (*e.g.*, Figure 2a).

**Self-Training** methods [17, 3, 53] are the mainstream in semi-supervised learning (SSL), and recently became a promising alternative in UDA, which focuses on generating accurate $T$ pseudo-labels. This lineup explores different classifier design, such as k-NN [51], a teacher-student network [11] or an exponential moving average model [30]. Yet the pseudo-labels are error-prune due to the spurious correlations in $S$, hence the model performance is still inconsistent [32].

**Alleviating Spurious Correlation** has been studied in several existing works of UDA [56, 32, 29]. They train a $T$-specific classifier to generate pseudo-labels [56, 29]. However, their $T$-classifiers are still initialized based on a classifier trained in $S$, hence the spurious correlation in $S$ are introduced to the $T$-classifiers right in the beginning. On the contrary, our ICON learns a cluster head in $T$ without supervision in $S$, which is solely for discovering the sample clusters in $T$. Hence, the spurious correlation in $S$ inconsistent with the clusters in $T$ can be removed by ICON (Section 5).

**Graph-based SSL** methods [16, 42, 22] share a similar spirit with ICON, where they learn a classifier using the labeled samples that is consistent with the unlabelled sample similarities. However, they lack the invariant constraint in ICON, which can cause the failure cases (*e.g.*, Figure 2c).

| Dataset | OFFICE-HOME | VISDA-2017 | IWILDCAM | CAMELYON17 | FMOW | POVERTYMAP | GLOBALWHEAT | OGB-MOLPCBA | CIVILCOMMENTS | AMAZON |
|---|---|---|---|---|---|---|---|---|---|---|
| Sample x | object image | object image | camera trap photo | tissue slide | satellite image | satellite image | wheat image | molecular graph | online comment | product review |
| Label y | 65 categories | 12 categories | 182 species | tumor/not | 62 land uses | asset wealth | wheat bbox | bioassays | toxic/not | 5 review scores |
| Task | classification | classification | classification | classification | classification | regression | detection | classification | classification | classification |
| Source S | various* | synthetic images | photos from 243 traps | slides from hospital 1-3 | images from 2002-2013 | images in 14 countries | images in Europe | 44,930 scaffold groups | online articles* | 1,252 reviewers |
| Example S | | | | | | | | | I applaud your father. He was a good man! We need more like him. | Super easy to put together. Very well built. |
| #Samples S | average 3,875 | 152,397 | 129,809 | 302,436 | 76,863 | ~10,000 | 2,943 | 350,343 | 269,038 | 245,502 |
| Target T | various* | real photos | photos from 3215 traps | slides from hospital 5 | images from 2016-2018 | images in 5 countries | images across the world | 43,793 scaffold groups | online articles* | 1,334 reviewers |
| Example T | | | | | | | | | As a Christian, I will not be patronizing any of those businesses. | I am disappointed in the quality of these. |
| #Samples T | average 3,875 | 55,388 | 819,120 | 600,030 | 173,208 | 261,396 | 42,445 | 517,048 | 1,551,515 | 268,761 |
| Evaluation | average acc. | mean-class accuracy | macro-F1 | acc. | worst-region acc.* | Pearson correlation* % | acc. | average precision | worst-group acc.* | 10th percentile acc. |

**Existing Methods**

| | OFFICE-HOME | VISDA-2017 | IWILDCAM | CAMELYON17 | FMOW | POVERTYMAP | GLOBALWHEAT | OGB-MOLPCBA | CIVILCOMMENTS | AMAZON |
|---|---|---|---|---|---|---|---|---|---|---|
| GVB [7] | | | Empirical Risk Minimization (ERM) | | | | | | | |
| | 70.4 | 75.3 | 47.0 / 32.2 | 90.6 / 82.0 | 60.6 / 34.8 | 65 / 48 | 77.8 / 51.0 | - / 28.3 | 89.8 / 66.6 | 72.0 / 54.2 |
| TCM [61] | | | CORAL [51] | | | | | | | |
| | 70.7 | 75.8 | 40.5 / 27.9 | 90.4 / 77.9 | 58.9 / 34.1 | 54 / 36 | - / - | - / 26.6 | - / - | 71.7 / 53.3 |
| SENTRY [41] | | | DANN [13] | | | | | | | |
| | 72.0 | 76.7 | 48.5 / 31.9 | 86.9 / 68.4 | 57.9 / 34.6 | 50 / 33 | - / - | - / 20.4 | - / - | 71.7 / 53.3 |
| CST [29] | | | Pseudo-Label [24] | | | | | | | |
| | 72.2 | 80.6 | 47.3 / 30.3 | 91.3 / 67.7 | 60.9 / 33.7 | - / - | 73.3 / 42.9 | - / 19.7 | 90.3 / 66.9 | 71.6 / 52.3 |
| ToAlign [59] | MDD [63] | | Noisy Student [60] | | | | | | | |
| | 72.7 | 77.8 | 47.5 / 32.1 | 93.2 / 86.7 | 61.3 / 37.8 | 61 / 42 | 78.1 / 46.8 | - / 27.5 | - / - | - / - |
| FixBi [37] | MT+16augs [11] | | FixMatch [50] | | | | | | Masked LM [9] | |
| | 73.0 | 82.8 | 46.3 / 31.0 | 91.3 / 71.0 | 58.6 / 32.1 | 54 / 30 | - / - | - / - | 89.4 / 65.7 | 71.9 / 53.9 |
| ATDOC [26] | MCC+NWD [6] | | SwAV [5] | | | | | | ERM (labelled T) | |
| | 73.2 | 83.7 | 47.3 / 29.0 | 92.3 / 91.4 | 61.8 / 36.3 | 60 / 45 | - / - | - / - | 89.9 / 69.4 | 73.6 / 56.4 |
| **ICON** | **75.8** | **87.4** | **50.6 / 34.5** | **95.6 / 93.8** | **62.2 / 39.9** | **65 / 49** | **78.6 / 52.3** | **- / 28.3** | **89.7 / 68.8** | **71.9 / 54.7** |
| | +2.6 | +3.7 | +2.3 | +2.4 | +2.1 | +1 | +1.3 | +0.0 | +1.9 | +0.5 |

Table 1: Dataset details and the results of our ICON compared with existing methods. Details on * in Section 5.1. WILDS 2.0 [47] datasets are highlighted in yellow, where models were evaluated on both the test data in $S$ (first number) and in $T$ (second number). The performance gain in $T$ highlighted in red. "ERM (labeled $T$)" has full supervision in $T$. -/- means that the method is not applicable to the dataset. Other dataset details and the standard deviation of the results are in Appendix.

## 5 Experiment

### 5.1 Datasets

As shown in Table 1, we extensively evaluated ICON on 10 datasets with standard protocols [34, 47], including 2 classic ones: OFFICE-HOME [60], VISDA-2017 [43], and 8 others from the recent WILDS 2.0 benchmark [47]. WILDS 2.0 offers large-scale UDA tasks in 3 modalities (image, text, graph) under real-world settings, *e.g.*, wildlife conservation, medical imaging and remote sensing. Notably, unlike the classic datasets where the same sample set in $T$ is used in training (unlabelled) and testing, WILDS 2.0 provides unlabelled train set in $T$ disjoint with the test set, which prevents explicit pattern mining of the test data. They also provide validation sets for model selection. For datasets in WILDS 2.0 benchmark, we drop their suffix -WILDS for simplicity (*e.g.*, denoting AMAZON-WILDS as AMAZON).

Details of * in Table 1: **OFFICE-HOME** has 4 domains: artistic images (Ar), clip art (Cl), product images (Pr), and real-world images (Rw), which forms 12 UDA tasks by permutation, and we only show Cl→Rw as an example in the table. **FMOW** considers the worst-case performance on different geographic regions, where the land use in Africa is significantly different. **POVERTYMAP** considers the lower of the Pearson correlations (%) on the urban or rural areas. **CIVILCOMMENTS** correspond to a semi-supervised setting with no domain gap between $S$ and $T$, and is evaluated on the worst-case performance among demographic identities.

On some WILDS 2.0 datasets, the test data may come from a separate domain different from the training target domain $T$, *e.g.*, IWILDCAM test data is captured by 48 camera traps disjoint from the 3215 camera traps in the training $T$. We highlight that ICON is still applicable on these datasets. This is because ICON works by learning the causal feature **c** and discarding the environmental feature **e**, and the theory holds as long as the definition of **c** and **e** is consistent across the training and test data, *e.g.*, IWILDCAM is about animal classification across the labeled, unlabeled, and test data.

| Method | Ar→Cl | Ar→Pr | Ar→Rw | Cl→Ar | Cl→Pr | Cl→Rw | Pr→Ar | Pr→Cl | Pr→Rw | Rw→Ar | Rw→Cl | Rw→Pr | Avg |
|---|---|---|---|---|---|---|---|---|---|---|---|---|---|
| DANN [13] (2016) | 45.6 | 59.3 | 70.1 | 47.0 | 58.5 | 60.9 | 46.1 | 43.7 | 68.5 | 63.2 | 51.8 | 76.8 | 57.6 |
| CDAN [34] (2018) | 50.7 | 70.6 | 76.0 | 57.6 | 70.0 | 70.0 | 57.4 | 50.9 | 77.3 | 70.9 | 56.7 | 81.6 | 65.8 |
| SymNet [69] (2019) | 47.7 | 72.9 | 78.5 | 64.2 | 71.3 | 74.2 | 64.2 | 48.8 | 79.5 | 74.5 | 52.6 | 82.7 | 67.6 |
| MDD [70] (2019) | 54.9 | 73.7 | 77.8 | 60.0 | 71.4 | 71.8 | 61.2 | 53.6 | 78.1 | 72.5 | 60.2 | 82.3 | 68.1 |
| SHOT[28] (2020) | 57.1 | 78.1 | 81.5 | 68.0 | 78.2 | 78.1 | 67.4 | 54.9 | 82.2 | 73.3 | 58.8 | 84.3 | 71.8 |
| ALDA [7] (2020) | 53.7 | 70.1 | 76.4 | 60.2 | 72.6 | 71.5 | 56.8 | 51.9 | 77.1 | 70.2 | 56.3 | 82.1 | 66.6 |
| GVB [8] (2020) | 57.0 | 74.7 | 79.8 | 64.6 | 74.1 | 74.6 | 65.2 | 55.1 | 81.0 | 74.6 | 59.7 | 84.3 | 70.4 |
| TCM [67] (2021) | 58.6 | 74.4 | 79.6 | 64.5 | 74.0 | 75.1 | 64.6 | 56.2 | 80.9 | 74.6 | 60.7 | 84.7 | 70.7 |
| SENTRY [44] (2021) | 61.8 | 77.4 | 80.1 | 66.3 | 71.6 | 74.7 | 66.8 | **63.0** | 80.9 | 74.0 | 66.3 | 84.1 | 72.2 |
| CST [32] (2021) | 59.0 | 79.6 | 83.4 | 68.4 | 77.1 | 76.7 | 68.9 | 56.4 | 83.0 | 75.3 | 62.2 | 85.1 | 73.0 |
| ToAlign [64] (2021) | 57.9 | 76.9 | 80.8 | 66.7 | 75.6 | 77.0 | 67.8 | 57.0 | 82.5 | 75.1 | 60.0 | 84.9 | 72.0 |
| FixBi [40] (2021) | 58.1 | 77.3 | 80.4 | 67.7 | 79.5 | 78.1 | 65.8 | 57.9 | 81.7 | **76.4** | 62.9 | 86.7 | 72.7 |
| ATDOC [29] (2021) | 60.2 | 77.8 | 82.2 | 68.5 | 78.6 | 77.9 | 68.4 | 58.4 | 83.1 | 74.8 | 61.5 | 87.2 | 73.2 |
| SDAT [45] (2022) | 58.2 | 77.1 | 82.2 | 66.3 | 77.6 | 76.8 | 63.3 | 57.0 | 82.2 | 74.9 | 64.7 | 86.0 | 72.2 |
| MCC+NWD [6] (2022) | 58.1 | 79.6 | 83.7 | 67.7 | 77.9 | 78.7 | 66.8 | 56.0 | 81.9 | 73.9 | 60.9 | 86.1 | 72.6 |
| kSHOT* [55] (2022) | 58.2 | 80.0 | 82.9 | 61.1 | 80.3 | 80.7 | **71.3** | 56.8 | 83.2 | 75.5 | 60.3 | 86.6 | 73.9 |
| **ICON (Ours)** | **63.3** | **81.3** | **84.5** | **70.3** | **82.1** | **81.0** | 70.3 | 61.8 | **83.7** | 75.6 | **68.6** | **87.3** | **75.8** |

Table 2: Break-down of the accuracies in each domain on OFFICE-HOME dataset [60]. *: kSHOT [55] additionally uses the prior knowledge on the percentage of samples in each class in the testing data. Published years are in the brackets after the method names.

## 5.2 Implementation Details

**Feature Backbone**. We used the followings pre-trained on ImageNet [46]: ResNet-50 [19] on OFFICE-HOME, VISDA-2017 and IWILDCAM, DenseNet-121 [21] on FMOW and Faster-RCNN [14] on GLOBALWHEAT. We used DistilBERT [48] with pre-trained weights from the Transformers library on CIVILCOMMENTS and AMAZON. On CAMELYON17, we used DenseNet-121 [21] pre-trained by the self-supervised SwAV [5] with the training data in $S$ and $T$. On POVERTYMAP and OGB-MOLPCBA with no pre-training available, we used multi-spectral ResNet-18 [19] and graph isomorphism network [66] trained with the labelled samples in the source domain $S$, respectively.

**Self-training**. A general form of the self-training loss $\mathcal{L}_{st}$ in Eq. (3) is given below:

$$\mathcal{L}_{st} = \mathbb{E}_{\mathbf{x} \sim T} \left[ \mathbb{1}(\max f(\mathbf{x}) > \tau) \mathrm{CE} \left( f(\mathbf{x}), \arg\max f(\mathbf{x}) \right) \right], \quad (4)$$

where $\tau$ is a confident threshold that discards low-confidence pseudo-labels, and $\mathrm{CE}(\cdot, \cdot)$ is the cross-entropy loss. We detail the different implementations of Eq (4) in Appendix. ICON is agnostic to the choice of $\mathcal{L}_{st}$, and hence we choose the best performing self-training baseline for each dataset: FixMatch [53] on OFFICE-HOME and VISDA-2017, NoisyStudent [65] on IWILDCAM and FMOW, Pseudo-label [26] on CIVILCOMMENTS and AMAZON. ICON also does not rely on $\mathcal{L}_{st}$. We did not apply $\mathcal{L}_{st}$ (*i.e.*, $\beta = 0$) on CAMELYON17, POVERTYMAP, GLOBALWHEAT and OGB-MOLPCBA where self-training does not work well.

**Regression and Detection**. For the regression task in POVERTYMAP, $f$ is a regressor that outputs a real number. As the number of clusters in $T$ is not well-defined, we directly used rank statistics to determine if two samples in $T$ have the same label. For the BCE loss, we computed the similarity by replacing $f(\mathbf{x}_i)^{\intercal} f(\mathbf{x}_j)$ in Eq. (1) as $-\left( f(\mathbf{x}_i) - f(\mathbf{x}_j) \right)^2$. For detection in GLOBALWHEAT, we used the ICON objective in Eq. (3) to train the classification head without other modifications.

**Other Details**. We aim to provide a simple yet effective UDA method that works across data modalities, hence we did not use tricks tailored for a specific modality (*e.g.*, image), such as mixup [68] or extensive augmentations (16 are used in [11]). On classic UDA datasets, we used entropy minimization and SAM optimizer following CST [32]. On VISDA-2017, IWILDCAM, CAMELYON17 and FMOW, we pre-processed the features with UMAP [37] (output dimension as 50) and EqInv [61], which helps fulfill Assumption 1 (*i.e.*, highlight causal feature to improve clustering). ICON is also resource efficient—all our experiments can run on a single NVIDIA 2080Ti GPU.

## 5.3 Main Results

**Classic UDA Benchmarks**. In Table 1 (first 2 columns), ICON significantly outperforms the existing state-of-the-arts on OFFICE-HOME [60] and VISDA-2017 [43] by 2.6% and 3.7%, respectively. We include the breaks down of the performances of OFFICE-HOME in Table 2, where ICON wins in 10 out of 12 tasks, and significantly improves the hard tasks (*e.g.*, Ar→Cl, Rw→Cl). For VISDA-2017, we include more comparisons in Table 3. ICON beats the original competition winner (MT+16augs) [11] that averages predictions on 16 curated augmented views. ICON even outperforms all methods that use the much deeper ResNet-101 [19]. Notably, CST [32] and ATDOC [29] also aim to remove

| Method | Backbone | Acc. |
|---|---|---|
| MT+16augs [11] (2018) | ResNet-50 | 82.8 |
| MDD [70] (2019) | ResNet-50 | 77.8 |
| GVB [8] (2020) | ResNet-50 | 75.3 |
| TCM [67] (2021) | ResNet-50 | 75.8 |
| SENTRY [44] (2021) | ResNet-50 | 76.7 |
| CST [32] (2021) | ResNet-50 | 80.6 |
| CAN [24] (2019) | ResNet-101 | 87.2 |
| SHOT [28] (2020) | ResNet-101 | 82.9 |
| FixBi [24] (2021) | ResNet-101 | 87.2 |
| MCC+NWD [6] (2022) | ResNet-101 | 83.7 |
| SDAC [45] (2022) | ResNet-101 | 84.3 |
| kSHOT* [55] (2022) | ResNet-101 | 86.1 |
| **ICON (Ours)** | ResNet-50 | **87.4** |

Table 3: Mean-class accuracy (Acc.) on VISDA-2017 Synthetic→Real task with the choice of feature backbone. *: details in Table 2 caption. Published years are in the brackets after the method names.

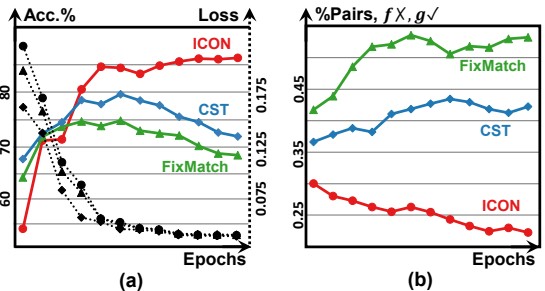

(a)          (b)

Figure 5: (a) Solid lines: test accuracy against training epochs for FixMatch, CST and ICON. Dotted black lines: average loss in each epoch with matching markers to denote the three methods. (b) Among the feature pairs where $f$ fails, the percentage of pairs where $g$ succeeds for the three methods.

the spurious correlation, and both attain competitive performance, validating the importance of this research objective. Yet their training scheme allows the correlation to persist (Section 4), hence ICON is more effective.

**WILDS 2.0**. In Table 1 (last 8 columns), ICON dominates the WILDS 2.0 leaderboard despite the challenging real-world data and stringent evaluation. For example, the animal images in IWILDCAM exhibit long-tail distribution that is domain-specific, because the rarity of species varies across locations. Hence under macro-F1 evaluation (sensitive to tail performance), the 2.3% improvement is noteworthy. On CIVILCOMMENTS with text data, ICON improves the worst performance across demographic groups by 2.2%, which promotes AI fairness and demonstrates the strength of ICON even under the SSL setting. On AMAZON, although ICON only improves 0.5% to reach 54.7%, the upper bound performance by training with labelled $T$ (56.4%) is also low due to the strict evaluation metric. We also highlight that ICON increases the performance in $T$ without sacrificing that in $S$ (the first number). This empirically validates that the invariance objective of ICON resolves the tug-of-war scenario in Figure 2c. In particular, we tried implementing recent methods on this benchmark. Yet they are outperformed by ERM, or even not applicable (*e.g.*, on text/graph data). This is in line with WILDS 2.0 organizers' observation that ERM as a naive baseline usually has the best performance, which is exactly the point of WILDS 2.0 as a diagnosis of the ever-overlooked issues in UDA. Hence we believe that dominating this challenging leaderboard is a strong justification for ICON, and encourage more researchers to try this benchmark.

**Failure Cases**. ICON performance is relatively weak on POVERTYMAP and OGB-MOLPCBA with two possible reasons: 1) no pre-training available, the backbone is trained only on $S$ and hence biased in the beginning, and 2) the ground-truth number of classes in $T$ is not well-defined in both datasets, hence Assumption 1 in Section 3 may fail.

**Learning Curve**. In Figure 5a, we compare our ICON with two self-training baselines: FixMatch [53] and CST [32]. The test accuracy and training loss *w.r.t.* training epochs on VISDA-2017 is drawn in colored solid lines and black dotted lines, respectively. We observe that the training losses for all methods converge quickly. However, while the two baselines achieve higher accuracy in the beginning, their performances soon start to drop. In contrast, ICON's accuracy grows almost steadily.

**Effect of Supervision in** $T$. To find the cause of the performance drop, we used the optimized $g$ in Section 2.2 to probe the learned $f$ in each epoch. Specifically, we sampled feature pairs $\mathbf{x}_1, \mathbf{x}_2$ in $T$ and tested if the predictions of $f, g$ are consistent with their labels, *e.g.*, $f$ succeeds if $\arg\max f(\mathbf{x}_1) = \arg\max f(\mathbf{x}_2)$ when $y_1 = y_2$ (vice versa for $\neq$). Then among the pairs where $f$ failed, we computed the percentage of them where $g$ succeeded, which is shown in Figure 5b. A large percentage implies that $f$ disrespects the inherent distribution in $T$ correctly identified by $g$. We observe that the percentage has a rising trend for the two baselines, *i.e.*, they become more biased to the spurious correlations in $S$ that do not hold in $T$. This also explains their dropping performance in Figure 5a. In particular, CST has a lower percentage compared to FixMatch, which suggests that it has some effects in removing the bias. In contrast, the percentage steadily drops for ICON, as we train $f$ to respect the inherent distribution identified by $g$.

| Method | OFFICE-HOME | VISDA-2017 |
|---|---|---|
| FixMatch | 69.1 | 76.6 |
| FixMatch+CON | 74.1 | 82.0 |
| FixMatch+CON+INV | 75.8 | 87.4 |
| Cluster with 2×#classes | 69.7 | 78.6 |
| Cluster with 0.5×#classes | 67.5 | 76.2 |
| Cluster with k-NN | 72.1 | 85.6 |

Table 4: Ablations on each ICON component. CON denotes the consistency loss in $S$ and $T$. INV denotes the invariance constraint.

| OFFICE-HOME | | | | | VISDA-2017 | | | | |
|---|---|---|---|---|---|---|---|---|---|
| $\beta$\$\alpha$ | 0.25 | 0.5 | 1.0 | 2.0 | $\beta$\$\alpha$ | 0.25 | 0.5 | 1.0 | 2.0 |
| 0.05 | 73.8 | 75.0 | 75.5 | 73.9 | 0.05 | 85.2 | 86.2 | 86.3 | 81.5 |
| 0.1 | 73.9 | 75.4 | 75.8 | 74.4 | 0.1 | 85.6 | 86.4 | 86.7 | 82.6 |
| 0.25 | 70.2 | 72.8 | 75.2 | 74.5 | 0.25 | 76.8 | 81.2 | 87.4 | 86.3 |

Figure 6: Ablations of $\alpha, \beta$ on the classic UDA datasets.

## 5.4 Ablations

**Components**. Recall that the main components of ICON include: 1) Consistency loss (CON) that leverages the supervision from the labels in $S$ and clusters in $T$; 2) the use of invariance constraint (INV) in Eq. (3); 2) clustering in $T$ with #clusters=#classes. Their ablations are included in Table 4. From the first three lines, we observe that adding each component leads to consistent performance increase on top of the self-training baseline (FixMatch), which validates their effectiveness. From the next two lines, when #clusters≠#classes, the model performs poorly, *e.g.*, using one cluster in Figure 2b (*i.e.*, 0.5×#classes), the invariant classifier needs to predict the two red circles in $S$ as dissimilar (two classes) and the two blue ones in $T$ as similar (one cluster), which in fact corresponds to the erroneous red line (*i.e.*, predictions in $T$ are similarly ambiguous). In the last line, we tried k-means for clustering instead of rank-statistics, leading to decreased performance. We postulate that rank-statistics is better because its online clustering provides up-to-date cluster assignments. In the future, we will try other more recent online clustering methods, *e.g.*, an optimal transport-based method [5]. Overall, the empirical findings validate the importance of the three components, which are consistent with the theoretical analysis in Section 3.

**Hyperparameters**. We used two hyperparameters in our experiments: 1) the weight of the self-training loss denoted as $\alpha$; 2) the invariance penalty strength $\beta$ in the REx [25] implementation of Eq. (3). Their ablations are in Figure 6 on the widely used classic UDA datasets. We observe that $\alpha \in [0.5, 1.0]$ and $\beta \in [0.1, 0.25]$ works well on both datasets. In particular, it is standard in the IRM community to choose a small $\beta$ (*e.g.*, 0.1). Then, tuning $\alpha$ follows the same process as other self-training UDA methods. Hence the hyperparameter search is easy.

**Confusion Matrix**. We use the matrix to concretely show how existing methods are biased to the correlations in the source domain $S$. We compare the state-of-the-art self-training method CST with ICON in Figure 7. The matrix on the left reveals that CST predicted most "knife"s as "skateboard"s ("board" for short), and some "truck"s as "car"s. On the right, we see that ICON predicts "knife" accurately and also improves the accuracy on "truck".

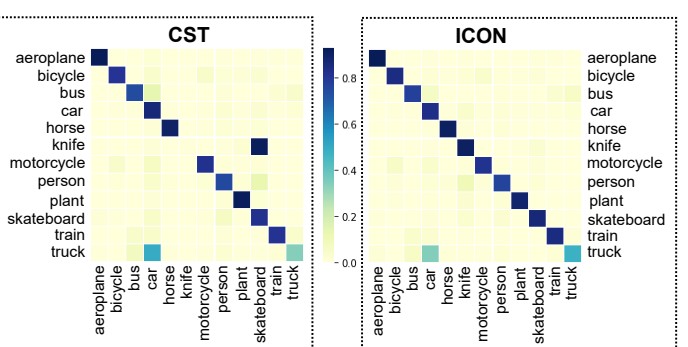

Figure 7: CST and ICON confusion matrix on VISDA-2017.

**Grad-CAM Visualization**. Why does CST predict "knife" in $T$ as "board"? To answer this, in Figure 8, we visualized the GradCAM [50] of CST and ICON trained on VisDA-2017. In the first row, we notice that the self-training based CST predicts the two categories perfectly by leveraging their apparent differences: the curved blade versus the flat board ending with semi-circles. However, this leads to poor generalization on "knife" in $T$, where the background (image 1, 4), the knife component (2) or even the blade (3) can have board-like appearances. Moreover, biased to the standard "board" look in $S$, CST can leverage environmental feature for predicting "board" in $T$, *e.g.*, under side-view (5) or slight obstruction (6), which is prune to errors (*e.g.*, 8). As the board-like shape alone is non-discriminative for the two categories in $T$, in the second row, the cluster head $g$ additionally leverages knife handles and wheels to distinguish them. In the third row, our ICON combines the

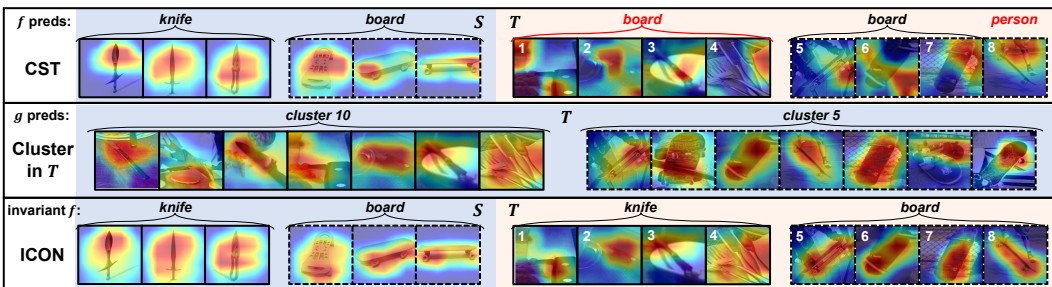

Figure 8: GradCAM on VisDA-2017 [43] using CST [32], the cluster head $g$ and the proposed ICON. Ground-truth knives and boards are in solid and dotted boxes, respectively. Predictions from $f, g$ are on top of each row. Red denotes wrong predictions. Pale blue and red background denotes training and testing, respectively. Note that $g$ is trained without supervision in $T$.

supervision in $S$ (first row) and $T$ (second row) to learn an invariant classifier $f$. In $S$, ICON focuses on the overall knife shape instead of just blade, and incorporates wheels to predict "board". In $T$, by comparing images with the same number in the first and third row, we see that ICON focuses more on the discriminative parts of the objects (*e.g.*, handles and wheels), hence generalizes better in $T$.

## 6 Conclusion

We presented a novel UDA method called Invariant CONsistency learning (ICON) that removes the spurious correlation specific to the source domain $S$ to generalize in the target domain $T$. ICON pursues an invariant classifier that is simultaneously consistent with the labels in $S$ and the clusters in $T$, hence removing the spurious correlation inconsistent in $T$. We show that ICON achieves generalization in $T$ with a theoretical guarantee. Our theory is verified by extensive evaluations on both the classic UDA benchmarks and the challenging WILDS 2.0 benchmark, where ICON outperforms all the conventional methods. We will seek more principled pre-training paradigms that disentangle the causal features and relax the assumption of knowing the class numbers in $T$ to improve regression tasks.

**Limitation**. Our approach is based on the assumptions in Section 4, *i.e.*, the classes in $T$ are separated by clusters, and there is enough diversity in $T$ such that causal feature **c** is the only invariance. Assumption 1 is essentially the clustering assumption, which is a necessary assumption for learning with unlabeled data [59]. In fact, UDA methods (and semi-supervised learning in general) commonly leverage feature pre-training and data augmentations to help fulfill the assumptions. One can also use additional pre-trained knowledge from foundation models, or deploy improved clustering algorithm (*e.g.*, an optimal transport-based method [5]). Assumption 2 is necessary to rule out corner cases, *e.g.*, in Figure 4b, without additional prior, there is no way for any UDA method to tell if each one of the blue cluster (unlabeled) should be pseudo-labeled as "0" or "1". In practice, these corner cases can be avoided by collecting diverse unlabeled data, which is generally abundant, *e.g.*, collecting unlabeled street images from a camera-equipped vehicle cruise.

**Broader Impact**. Our work aims to capture the causal feature in the unsupervised domain adaptation setting. In particular, we significantly improve the state-of-the-art on WILDS 2.0 benchmark, which includes practical tasks that benefit wildlife preservation, tumour classification in healthcare, remote sensing, vision in agriculture, *etc*. The pursue of causal feature can produce more robust, transparent and explainable models, broadening the applicability of ML models and promoting fairness in AI.

## 7 Acknowledgement

This research is supported by the National Research Foundation, Singapore under its AI Singapore Programme (AISG Award No: AISG2-RP-2021-022), MOE AcRF Tier 2 (MOE2019-T2-2-062), Alibaba-NTU Singapore Joint Research Institute, the A*STAR under its AME YIRG Grant (Project No.A20E6c0101), and the Lee Kong Chian (LKC) Fellowship fund awarded by Singapore Management University.

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
