# Appendix

This appendix is organized as follows:

- Section A provides more details about our algorithms. In particular, Section A.1 describes the rank-statistics clustering algorithm. Section A.2 describes the REx implementation of invariance in Eq. (2). Section A.3 discusses the two implementation of the self-training used in our experiments: FixMatch [16] and Noisy Student [19].
- Section B gives the preliminaries of group theory.
- Section C gives the full proof of our theorem.
- Section D provides more details on the datasets, our feature backbone, how we use seeds in the evaluation, self-training details, and other implementation details.
- Section E shows more comparisons and standard deviations on WILDS 2.0 benchmark [14], VISDA-2017 [12], and the break-down of accuracy in each domain for OFFICE-HOME [17].
- Codes are also provided, which include the training and testing scripts on the two classic datasets, WILDS 2.0 benchmark, and the live demo code for Figure 2 using MNIST [3]. The setup instructions and commands used in our experiments are included in the `README.md` file.

| Abbreviation/Symbol | Meaning |
|---|---|
| *Abbreviation* | |
| DA | Domain Adaptation |
| UDA | Unsupervised Domain Adaptation |
| BCE | Binary Cross-Entropy |
| CE | Cross-Entropy |
| *Symbol in Algorithm* | |
| $S$ | Source domain |
| $T$ | Target domain |
| $\mathbf{c}$ | Causal feature |
| $\mathbf{e}$ | Environmental feature |
| $\Phi$ | (Hidden) Data generator |
| $\mathbf{x}$ | Sample feature |
| $y$ | Sample label |
| $\theta$ | Parameters of the backbone |
| $N, M$ | Number of samples in $S, T$, respectively |
| $f$ | Classification head |
| $g$ | Cluster head |
| $\mathcal{L}_{st}$ | Self-training loss |
| $\alpha$ | Self-training loss weight |
| $\beta$ | Invariance loss weight |
| *Symbol in Theory* | |
| $\mathcal{G}$ | Group of semantics |
| $\mathcal{X}$ | Feature space |
| $\mathcal{H}$ | Subgroup of $\mathcal{G}$ that transforms environmental feature $\mathbf{e}$ |
| $g \circ \mathbf{x}$ | Group action |

Table A1: List of abbreviations and symbols used in the paper.

# A Additional Details on Algorithm

## A.1 Rank-Statistics Clustering

Recall that in UDA training, a set of $M$ unlabelled sample features $\{\mathbf{x}_i\}_{i=1}^M$ (features extracted by the backbone $\theta$) from $C$ classes is available in the target domain $T$. Rank-statistics clustering algorithm [5] learns a cluster head $g$, whose output $g(\mathbf{x}_i)$ are the softmax-normalized probabilities for $\mathbf{x}_i$ belonging to each of the $C$ clusters. We use the pair-wise binary cross-entropy (BCE) loss to learn $g$:

$$
\min_g \mathbb{E}_{\mathbf{x}_i, \mathbf{x}_j \in T} - y_{i,j}\log\left(g(\mathbf{x}_i)^\mathsf{T} g(\mathbf{x}_j)\right) \\
- (1 - y_{i,j})\log\left(1 - g(\mathbf{x}_i)^\mathsf{T} g(\mathbf{x}_j)\right),
\tag{A1}
$$

where $y_{i,j} = 1$ if the top-5 values of $\mathbf{x}_i, \mathbf{x}_j$ have the same indices, and 0 otherwise. In practice, the above optimization can be implemented efficiently in a batch-wise sampling manner, where we compute the pair-wise BCE loss over each sample batch, instead of the entire unlabelled sample set in $T$.

## A.2 REx

REx circumvents the challenging bi-level optimization from line 2 of Eq. (2) by adding the following invariant constraint loss $\mathcal{L}_{REx}$ to line 1:

$$
\mathcal{L}_{REx} = \beta \mathrm{Var}\left(\{\mathrm{BCE}(S, f), \mathrm{BCE}(T, f)\}\right),
\tag{A2}
$$

which computes the variance between $\mathrm{BCE}(S, f)$ and $\mathrm{BCE}(T, f)$ with a trade-off parameter $\beta$. We performed ablations on $\beta$ in Section 5.4. Please refer to [8] for a theoretical explanation.

## A.3 Self-Training

We adopt three implementations of self-training, and use a trade-off parameter $\alpha$ to balance the weight of self-training loss.

**FixMatch** [16]. Given the classifier $f$ and a pair of features $\mathbf{x}, \mathbf{x}'$ corresponding to a sample's feature under weak and strong augmentations, respectively, the FixMatch self-training loss $\mathcal{L}_{fm}$ is given by:

$$
\mathcal{L}_{st} = \mathbb{1}(\mathrm{argmax} f(\mathbf{x}) > \tau)\mathrm{CE}(\mathbf{x}', \mathrm{argmax} f(\mathbf{x})),
\tag{A3}
$$

where $\mathbb{1}$ is the indicator function that returns 1 if and only if the condition in the bracket is true, $\tau$ is the confident threshold, and CE denotes the cross-entropy loss. In our experiments, we use the standard [10] fixed threshold $\tau = 0.97$ for the two classic datasets, and follow the standard threshold settings in WILDS 2.0 without searching. Hence we did not perform ablations on $\tau$.

**Noisy Student** [19]. Compared to FixMatch, Noisy Student has the following differences: 1) Instead of minimizing the self-training loss in an online fashion, Noisy Student generates the pseudo-labels $\mathrm{argmax} f(\mathbf{x})$ for all samples in $T$ at the start of epoch using the current model, and the model is updated for one epoch using the fixed pseudo-labels. 2) It does not perform selection by confident threshold (*i.e.*, $\mathbb{1}(\cdot)$ in Eq. (A3)), and hence the pseudo-labels for all samples participate in training. 3) Unlike FixMatch that relies on the use of strong augmentations to prevent model collapsing, Noisy Student can still be used when only weak augmentations are well-defined on the dataset, allowing it to run on more datasets (Table 1).

**Pseudo-Label** [9]. We adopt the implementation by WILDS 2.0 benchmark:

$$
\mathcal{L}_{st} = \mathbb{1}(\mathrm{argmax} f(\mathbf{x}') > \tau)\mathrm{CE}(\mathbf{x}', \mathrm{argmax} f(\mathbf{x}')),
\tag{A4}
$$

which uses only one augmented view of each sample, instead of the weak and strong augmented views in FixMatch. Note that the augmentation is optional in Eq. (A4), *e.g.*, on modalities where data augmentation is not available/difficult (*e.g.*, text data), one can use Pseudo-Label without augmentation. This implementation also linearly scales up the self-training weight from 0 to the specified $\alpha$ through the course of 40% total training steps.

# B  Group Theory Preliminaries

A group is a set together with a binary operation, which takes two elements in the group and maps them to another element. For example, the set of integers is a group under the binary operation of plus. We formalize the notion through the following definition.

**Binary Operation**. A binary operation $\cdot$ on a set $\mathcal{S}$ is a function mapping $\mathcal{S} \times \mathcal{S}$ into $\mathcal{S}$. For each $(s_1, s_2) \in \mathcal{S} \times \mathcal{S}$, we denote the element $\cdot(s_1, s_2)$ by $s_1 \cdot s_2$.

**Group**. A group $\langle \mathcal{G}, \cdot \rangle$ is a set $\mathcal{G}$, closed under a binary operation $\cdot$, such that the following axioms hold:

1. *Associativity.* $\forall g_1, g_2, g_3 \in \mathcal{G}$, we have $(g_1 \cdot g_2) \cdot g_3 = g_1 \cdot (g_2 \cdot g_3)$.
2. *Identity Element.* $\exists e \in \mathcal{G}$, such that $\forall g \in \mathcal{G}$, $e \cdot g = g \cdot e = g$.
3. *Inverse.* $\forall g \in \mathcal{G}$, $\exists g' \in \mathcal{G}$, such that $g \cdot g' = g' \cdot g = e$.

The binary operator $\cdot$ is often omitted, *i.e.*, we write $g_1 g_2$ instead of $g_1 \cdot g_2$. Groups often arise as transformations of some space, such as a set, vector space, or topological space. Consider an equilateral triangle. The set of clockwise rotations *w.r.t.* its centroid to retain its appearance forms a group $\{60°, 120°, 180°\}$, with the last element corresponding to an identity mapping. We say this group of rotations act on the triangle, formally defined below.

**Group Action**. Let $\mathcal{G}$ be a group and $\mathcal{S}$ be a set. An action of $\mathcal{G}$ on $\mathcal{S}$ is a map $\pi : \mathcal{G} \to \mathrm{Hom}(\mathcal{S}, \mathcal{S})$ so that $\pi(e) = \mathrm{id}_{\mathcal{S}}$ and $\pi(g)\pi(h) = \pi(gh)$, where $g, h \in \mathcal{G}$. $\forall g \in \mathcal{G}, s \in \mathcal{S}$, denote $\pi(g)(s)$ as $g \circ s$.

**Direct Product of Group**. Let $\mathcal{G}_1, \ldots, \mathcal{G}_n$ be groups with the binary operation $\cdot$. Let $a_i, b_i \in \mathcal{G}_i$ for $i \in \{1, \ldots, n\}$. Define $(a_1, \ldots, a_n) \cdot (b_1, \ldots, b_n)$ to be the element $(a_1 \cdot b_1, \ldots, a_n \cdot b_n)$. Then $\mathcal{G}_1 \times \ldots \times \mathcal{G}_n$ is the direct product of the groups $\mathcal{G}_1, \ldots, \mathcal{G}_n$ under the binary operation $\cdot$.

**Quotient Group**. Let $\phi : \mathcal{G} \to \mathcal{G}'$ mapping to some group $\mathcal{G}'$ be a group homomorphism, *i.e.*, $\phi(gh) = \phi(g)\phi(h)$, such that $\mathcal{H} = \{h \in \mathcal{G} \mid \phi(h) = e'\}$. Let $g \in \mathcal{G}$. Then the coset $a\mathcal{H}$ of $\mathcal{H}$ is defined as $\{g \in \mathcal{G} \mid \phi(g) = \phi(a)\}$, and the cosets of $\mathcal{H}$ form a quotient group $\mathcal{G}/\mathcal{H}$.

**Theorem**. Let $\mathcal{G} = \mathcal{K} \times \mathcal{H}$ be the direct product of group $\mathcal{K}$ and $\mathcal{H}$. Let $\bar{\mathcal{H}} = \{(e, h) \mid h \in \mathcal{H}\}$. Then $\mathcal{G}/\bar{\mathcal{H}}$ is isomorphic to $\mathcal{K}$ in a natural way.

In our formulation, we consider a semantic space $\mathcal{S} = \mathcal{C} \times \mathcal{E}$ as the Cartesian product of the causal feature space $\mathcal{C}$ and the environmental feature space $\mathcal{E}$. The group $\mathcal{G}$ acting on $\mathcal{S}$ corresponds to all semantic transformations, *e.g.*, $g \in \mathcal{G}$ may correspond to "turn darker" of the digit color. In particular, $\mathcal{G}$ has a corresponding direct product decomposition $\mathcal{G} = \mathcal{K} \times \mathcal{H}$, where $\mathcal{K}$ acts on $\mathcal{C}$ and $\mathcal{H}$ acts on $\mathcal{E}$. With slight abuse of notation, we write $\mathcal{K}$ as $\mathcal{G}/\mathcal{H}$ with the above theorem.

**$\mathcal{G}$-Equivariant Map**. Let $\mathcal{G}$ be a group acting on the set $\mathcal{S}$. Then $\Phi : \mathcal{S} \to \mathcal{X}$ mapping to the set $\mathcal{X}$ is said to be a $\mathcal{G}$-equivariant map if $f(g \circ \mathbf{s}) = g \circ f(\mathbf{s})$ for all $g \in \mathcal{G}, \mathbf{s} \in \mathcal{S}$.

In our work, we consider the mapping from the causal and environmental features to the learned sample features as a $\mathcal{G}$-equivariant map, hence we write $g \circ \mathbf{x}$ for $g \in \mathcal{G}, \mathbf{x} \in \mathcal{X}$ as the transformed feature from $\mathbf{x}$ by $g$.

**Group Representation**. Let $\mathcal{G}$ be a group. A representation of $\mathcal{G}$ (or $\mathcal{G}$-representation) is a pair $(\pi, \mathcal{X})$, where $\mathcal{X}$ is a vector space and $\pi : \mathcal{G} \to \mathrm{Hom}_{vect}(\mathcal{X}, \mathcal{X})$ is a group action, *i.e.*, for each $g \in \mathcal{G}, \pi(g) : \mathcal{X} \to \mathcal{X}$ is a linear map.

Intuitively, each $g \in \mathcal{G}$ corresponds to a linear map, *i.e.*, a matrix $\mathbf{M}_g$ that transforms a vector $\mathbf{x} \in \mathcal{X}$ to $\mathbf{M}_g \mathbf{x} \in \mathcal{X}$.

# C  Proof of Theorem

We include our assumptions and theorem in the main paper below for easy reference.

**Assumption** (Identifiability of $\mathcal{G}/\mathcal{H}$).
*1) $\forall \mathbf{x}_i, \mathbf{x}_j \in T$, $y_i = y_j$ iff $\mathbf{x}_i \in \{h \circ \mathbf{x}_j \mid h \in \mathcal{H}\}$;*
*2) There exists no linear map $l : \mathcal{X} \to \mathbb{R}$ such that $l(\mathbf{x}) > l(h \circ \mathbf{x})$, $\forall \mathbf{x} \in S, h \circ \mathbf{x} \in T$ and $l(g \circ \mathbf{x}) < l(gh \circ \mathbf{x})$, $\forall g \circ \mathbf{x} \in S, gh \circ \mathbf{x} \in T$, where $h \neq e \in \mathcal{H}, g \neq e \in \mathcal{G}/\mathcal{H}$.*

**Theorem**. *When the above assumptions hold, ICON (optimizing Eq. 2) learns a backbone $\theta$ mapping to a feature space $\mathcal{X}$ that generalizes under $\mathcal{G}/\mathcal{H} \times \mathcal{H}$. In particular, the learned $f$ is the optimal classifier in $S$ and $T$.*

*Proof sketch.* We will start by proving the binary classification case: we first show that the classifier satisfying the invariance objective in Eq. (2) (*i.e.*, line 2) separates the two classes among all samples in $S$ and $T$. Then by minimizing line 1, samples are mapped to two class-specific features (*i.e.*, one feature for each class). Finally, by minimizing CE using labelled samples in $S$, we learn the optimal classifier that assigns the correct labels to all samples in $S$ and $T$. We expand the binary classification case to $C$-class classification by formulating it as $C$ 1-versus-$(C-1)$ binary classification problems.

*Proof.* Consider a binary classification problem, where $f$ outputs a single probability of belong to the first class, $\mathcal{G}/\mathcal{H} = \{g, e\}$, *i.e.*, $\mathbf{x}, g \circ \mathbf{x}$ are from different classes for all $\mathbf{x} \in \mathcal{X}$. Let $f$ satisfies the invariance objective in Eq. (2), *i.e.*, $f \in \mathrm{argmin}_{\bar{f}}\mathrm{BCE}(S, \bar{f}) \cap \mathrm{argmin}_{\bar{f}}\mathrm{BCE}(T, \bar{f})$. As $\mathrm{BCE}(S, f)$ is computed using the ground-truth labels, we must have $f(\mathbf{x}_1) \to 1, f(\mathbf{x}_2) \to 0$ or $f(\mathbf{x}_1) \to 0, f(\mathbf{x}_2) \to 1, \ \forall \mathbf{x}_1 \in \mathcal{H}(\mathbf{x}) \cap S, \mathbf{x}_2 \in \mathcal{H}(g \circ \mathbf{x}) \cap S$, where $\mathbf{x} \in \mathcal{X}$. To see this, we shall expand $\mathrm{BCE}(S, \bar{f})$ on a pair of samples $\mathbf{x}, \mathbf{x}' \in S$ with label $y, y'$:

$$-\mathbb{1}(y = y')\log(\hat{\mathbf{y}}^{\mathsf{T}}\hat{\mathbf{y}}') - \mathbb{1}(y \neq y')\log(1 - \hat{\mathbf{y}}^{\mathsf{T}}\hat{\mathbf{y}}'), \tag{A5}$$

where $\mathbb{1}$ is the indicator function, and $\hat{\mathbf{y}}$ is given by $(f(\mathbf{x}), 1 - f(\mathbf{x}))$ (similarly for $\hat{\mathbf{y}}'$). Note that it is not difficult to show that $\hat{\mathbf{y}}^{\mathsf{T}}\hat{\mathbf{y}}' \in [0, 1]$, where the lower (or upper) bound is attained when they take different (or same) values from $(0, 1), (1, 0)$. As $f$ minimizes $\mathrm{BCE}(S, \bar{f})$, it minimizes the loss for each $\mathbf{x}, \mathbf{x}' \in S$: if $y = y'$, $\hat{\mathbf{y}}^{\mathsf{T}}\hat{\mathbf{y}}' \to 1$, and if $y \neq y'$, $\hat{\mathbf{y}}^{\mathsf{T}}\hat{\mathbf{y}}' \to 0$. This is only possible when the aforementioned condition holds. In the target domain $T$, although there is no ground-truth label, we prove that the evaluation of $\mathbb{1}(y = y')$ (or $\neq$) using the cluster labels is always the same as using ground-truth class labels, for all $\mathbf{x}, \mathbf{x}' \in T$ given Assumption 1: 1) suppose that $\mathbf{x} \in \{h \circ \mathbf{x}' \mid h \in \mathcal{H}\}$, then by the sufficient condition of Assumption 1, $\mathbb{1}(y = y') = 1$, reflecting that they are from the same class; 2) suppose that $\mathbf{x} \notin \{h \circ \mathbf{x}' \mid h \in \mathcal{H}\}$, then by the contra-position of the necessary condition, $\mathbb{1}(y \neq y') = 1$, reflecting that they are from the different classes. Hence in the same way as $S$, we can show that $f(\mathbf{x}_1) \to 1, f(\mathbf{x}_2) \to 0$ or $f(\mathbf{x}_1) \to 0, f(\mathbf{x}_2) \to 1, \ \forall \mathbf{x}_1 \in \mathcal{H}(\mathbf{x}) \cap T, \mathbf{x}_2 \in \mathcal{H}(g \circ \mathbf{x}) \cap T$. Now considering the samples in $S$ and $T$. Denote $D = S \cup T$. There are two possibilities: **1)** $f(\mathbf{x}_1) \to 1, f(\mathbf{x}_2) \to 0$ or $f(\mathbf{x}_1) \to 0, f(\mathbf{x}_2) \to 1, \ \forall \mathbf{x}_1 \in \mathcal{H}(\mathbf{x}) \cap D, \mathbf{x}_2 \in \mathcal{H}(g \circ \mathbf{x}) \cap D$; **2)** $f(\mathbf{x}_1) \to 1, f(\mathbf{x}_2) \to 0$ (or $f(\mathbf{x}_1) \to 0, f(\mathbf{x}_2) \to 1$), $\forall \mathbf{x}_1 \in \mathcal{H}(\mathbf{x}) \cap S, \mathbf{x}_2 \in \mathcal{H}(g \circ \mathbf{x}) \cap S$ and $f(\mathbf{x}_1) \to 0, f(\mathbf{x}_2) \to 1$ (or $f(\mathbf{x}_1) \to 0, f(\mathbf{x}_2) \to 1$), $\forall \mathbf{x}_1 \in \mathcal{H}(\mathbf{x}) \cap T, \mathbf{x}_2 \in \mathcal{H}(g \circ \mathbf{x}) \cap T$. Case 2 corresponds to the horizontal black line in Figure 4b. However, it contradicts with Assumption 2. Hence the only possibility is case 1, where $f$ linearly separates the two classes in $D$, *i.e.*, all samples in $S$ and $T$. In particular, as Assumption 2 prevents the failure case where $f$ separates the samples not based on the class (or causal feature $\mathbf{c}$), it intuitively means that the only invariance between $S$ and $T$ is $\mathbf{c}$. Overall, this shows that a linear map $f \in \mathrm{argmin}_{\bar{f}}\mathrm{BCE}(S, \bar{f}) \cap \mathrm{argmin}_{\bar{f}}\mathrm{BCE}(T, \bar{f})$ separates the two classes in $D$.

We will now analyze the general case of $c$-class classification. Specifically, $\mathbf{w} \in \mathbb{R}^{c \times d}$ corresponds to a linear map that transforms each sample feature $\mathbf{x} \in \mathbb{R}^d$ to the $c$-dimensional logits. The classifier $f$ parameterized by $\mathbf{w}$ outputs the softmax-normalized logits as the probability of belong to each of the $c$ class. Without loss of generality, we consider the case where $\|\mathbf{w}_i\| = \|\mathbf{w}_j\|$ for $i, j \in \{1, \dots, c\}$ and $\|\mathbf{x}\| = M$ for all $\mathbf{x} \in \mathcal{X}$, *e.g.*, through the common l2-regularization and feature normalization. From the previous analysis, we prove that a linear classifier satisfying the invariance objective separates the two classes in $D$. We can generalize the results to multi-class classification by considering it as $C$ 1-versus-$(C-1)$ binary classification problem (*e.g.*, separating class 1 samples and the rest of samples). We can further show that minimizing line 1 of Eq. (2) leads to a feature space $\mathcal{X}$ that generalizes under $\mathcal{G}/\mathcal{H} \times \mathcal{H}$. In particular, to minimize $\mathrm{BCE}(S, f) + \mathrm{BCE}(T, f)$ (expanded in Eq. (A5)), the prediction $p$ of $f$ on the correct class is maximized (*i.e.*, $p \to 1$). Specifically, the probability of belonging to class $i \in \{1, \dots, c\}$ for a feature $\mathbf{x}$ using softmax normalization is given by:

$$\frac{\exp(\mathbf{w}_i^{\mathsf{T}}\mathbf{x})}{\sum_{j=1}^{c} \exp(\mathbf{w}_j^{\mathsf{T}}\mathbf{x})}. \tag{A6}$$

We show that to maximize $p$, a sample feature in class $i$ is given by $M\bar{\mathbf{w}}_i$, where $M \to \infty$, $\bar{\mathbf{w}}_i$ is from normalizing $\mathbf{w}_i$, and for each $i \neq j \in \{1, \dots, c\}$, $\mathbf{w}_i^{\mathsf{T}}\mathbf{w}_j \leq 0$: the first condition is easily derived by maximizing $\exp(\mathbf{w}_i^{\mathsf{T}}\mathbf{x}) \to \infty$ under $\|\mathbf{x}\| = M$; the second condition is derived by limiting $p \to 1$, *i.e.*, $\exp(\mathbf{w}_j^{\mathsf{T}}\mathbf{x}) = 0$ or $\exp(\mathbf{w}_j^{\mathsf{T}}\mathbf{x}) \to -\infty$. Overall, this shows that minimizing Eq. (2) leads to $\mathcal{X}$

where the samples in class $i$ are mapped to the same feature $M\bar{\mathbf{w}}_i$, and samples in different classes are mapped to different features as $\mathbf{w}_i \neq \mathbf{w}_j$ for $i \neq j$ (from $\mathbf{w}_i^\intercal \mathbf{w}_j \leq 0$). The classifier $f$ is optimal by predicting the probability of each feature in $S$ and $T$ belonging to its ground-truth class as 1. Note that to further equip $\mathcal{X}$ with the equivariant property as introduced in Section **??**, one can leverage a specially designed backbone $\theta$ (*e.g.*, equivariant neural networks [2]). Hence we complete the proof of our Theorem.

## D  Dataset and Implementations

**Additional Dataset Details**. For datasets in WILDS 2.0 benchmark, we drop their suffix -WILDS for simplicity in the main text (*e.g.*, denoting AMAZON-WILDS as AMAZON). For IWILDCAM, the test data comes from a disjoint set of 48 camera traps (*i.e.*, different from the 3215 camera traps in the training $T$). For POVERTYMAP, each sample is a multi-spectral satelite image with 8 channels. OGB-MOLPCBA corresponds to a multi-task classification setting, where each label $y$ is a 128-dimensional binary vector, representing the binary outcomes of the biological assay results. $y$ could contain NaN values, *i.e.*, the corresponding biological assays were not performed on the given molecule. Hence there is no way to determine the ground-truth class numbers in this dataset. In AMAZON, $S$ and $T$ each contains a disjoint set of reviewers, and models are evaluated at their performance on the reviewer at the 10th percentile.

**Feature Backbone**. We used the followings pre-trained on ImageNet [13]: ResNet-50 [6] on 2 classic datasets and IWILDCAM, DenseNet-121 [7] on FMoW and Faster-RCNN [4] on GLOB-ALWHEAT. We used DistilBERT [15] with pre-trained weights from the Transformers library on CIVILCOMMENTS and AMAZON. On CAMELYON17, we used DenseNet-121 [7] pre-trained by the self-supervised SwAV [1] with the training data in $S$ and $T$. On POVERTYMAP and OGB-MOLPCBA with no pre-training available, we used multi-spectral ResNet-18 [6] and graph isomorphism network [20] trained with the labelled samples in the source domain $S$, respectively.

**Seeds**. We followed the standard evaluation protocol: POVERTYMAP provides 5 different cross-validation folds that split the dataset differently. Hence we used a single seed (0), and average the performances on the 5 folds. We evaluated our model on CAMELYON17 with 10 seeds (0-9), and CIVILCOMMENTS with 5 seeds (0-4). We used 3 seeds (0-2) on other datasets in WILDS 2.0. We fixed the seed for Python, NumPy and PyTorch as per standard.

**Self-Training Details**. We used FixMatch [16] on the two classic UDA datasets. On the WILDS 2.0 datasets with text data, we used pseudo-label method [9], which is similar to FixMatch, but without the use of weak and strong augmentation. On other WILDS 2.0 datasets, we used Noisy Student [19] due to its superior performance.

**Highlight causal feature c**. We use two techniques to highlight the causal feature **c**, such that Assumption 1 is better fulfilled: 1) On large scale datasets VISDA-2017 and IWILDCAM, we deployed UMAP [11] to reduce the dimension of $T$ features to 50 before clustering them. UMAP is empirically validated to faithfully capture the local data structure (*e.g.*, high k-NN classification accuracy) as well as the global structure (*e.g.*, more meaningful global relationships between different classes) in a low-dimensional space. Such structural information is however elusive in the original high-dimensional space due to the curse of dimensionality. 2) On image datasets VISDA-2017, IWILDCAM, CAMELYON and FMoW, we deploy EqInv [18] to further pursue the causal feature.

**Other Implementation Details**. For experiments on the classic UDA datasets, we modified the open-source code base of CST [10] (`https://github.com/Liuhong99/CST`). For experiments on WILDS 2.0 datasets, we implemented ICON on the official code base (`https://github.com/p-lambda/wilds`). On the classic UDA datasets, we followed the hyperparameter settings of CST except for the self-training weight, which we did an ablation in Figure 6. On WILDS 2.0 benchmark, we followed the released hyperparameter settings in their original experiments (`https://worksheets.codalab.org/worksheets/0x52cea64d1d3f4fa89de326b4e31aa50a`). In practice, we removed the $\text{BCE}(S, f)$ from line 1 of Eq. (2), as $\text{CE}(S, f)$ alone is enough to achieve consistency *w.r.t.* labels in $S$. We also selected sample pairs with confident binary label to compute $\text{BCE}(T, f)$ in line 1. Please refer to the attached code and the `README.md` for additional details.

| | IWILDCAM2020-WILDS | | FMOW-WILDS | |
| | (Unlabeled extra, macro F1) | | (Unlabeled target, worst-region acc) | |
| | In-distribution | Out-of-distribution | In-distribution | Out-of-distribution |
|---|---|---|---|---|
| ERM (-data aug) | 46.7 (0.6) | 30.6 (1.1) | 59.3 (0.7) | 33.7 (1.5) |
| ERM | 47.0 (1.4) | 32.2 (1.2) | 60.6 (0.6) | 34.8 (1.5) |
| CORAL | 40.5 (1.4) | 27.9 (0.4) | 59.3 (0.7) | 33.7 (1.5) |
| DANN | 48.5 (2.8) | 31.9 (1.4) | 57.9 (0.8) | 34.6 (1.7) |
| Pseudo-Label | 47.3 (0.4) | 30.3 (0.4) | 60.9 (0.5) | 33.7 (0.2) |
| FixMatch | 46.3 (0.5) | 31.0 (1.3) | 58.6 (2.4) | 32.1 (2.0) |
| Noisy Student | 47.5 (0.9) | 32.1 (0.7) | 61.3 (0.4) | 37.8 (0.6) |
| SwAV | 47.3 (1.4) | 29.0 (2.0) | 61.8 (1.0) | 36.3 (1.0) |
| ERM (fully-labeled) | 54.6 (1.5) | 44.0 (2.3) | 65.4 (0.4) | 58.7 (1.4) |
| **ICON (Ours)** | **50.6** (1.3) | **34.5** (1.4) | **62.2** (0.4) | **39.9** (1.1) |
| | CAMELYON17-WILDS | | POVERTYMAP-WILDS | |
| | (Unlabeled target, avg acc) | | (Unlabeled target, worst U/R corr) | |
| | In-distribution | Out-of-distribution | In-distribution | Out-of-distribution |
| ERM (-data aug) | 85.8 (1.9) | 70.8 (7.2) | 0.65 (0.03) | 0.48 (0.04) |
| ERM | 90.6 (1.2) | 82.0 (7.4) | **0.66** (0.04) | 0.48 (0.05) |
| CORAL | 90.4 (0.9) | 77.9 (6.6) | 0.54 (0.10) | 0.36 (0.08) |
| DANN | 86.9 (2.2) | 68.4 (9.2) | 0.50 (0.07) | 0.33 (0.10) |
| Pseudo-Label | 91.3 (1.3) | 67.7 (8.2) | - | - |
| FixMatch | 91.3 (1.1) | 71.0 (4.9) | 0.54 (0.11) | 0.30 (0.11) |
| Noisy Student | 93.2 (0.5) | 86.7 (1.7) | 0.61 (0.07) | 0.42 (0.11) |
| SwAV | 92.3 (0.4) | 91.4 (2.0) | 0.60 (0.13) | 0.45 (0.05) |
| **ICON (Ours)** | **95.6** (0.2) | **93.8** (0.3) | 0.65 (0.05) | **0.49** (0.04) |
| | GLOBALWHEAT-WILDS | | OGB-MOLPCBA | |
| | (Unlabeled target, avg domain acc) | | (Unlabeled target, average AP) | |
| | In-distribution | Out-of-distribution | In-distribution | Out-of-distribution |
| ERM | 77.8 (0.2) | 51.0 (0.7) | - | **28.3** (0.1) |
| CORAL | - | - | - | 26.6 (0.2) |
| DANN | - | - | - | 20.4 (0.8) |
| Pseudo-Label | 73.3 (0.9) | 42.9 (2.3) | - | 19.7 (0.1) |
| Noisy Student | 78.1 (0.3) | 46.8 (1.2) | - | 27.5 (0.1) |
| **ICON (Ours)** | **78.6** (0.0) | **52.3** (0.2) | - | **28.3** (0.0) |
| | CIVILCOMMENTS-WILDS | | AMAZON-WILDS | |
| | (Unlabeled extra, worst-group acc) | | (Unlabeled target, 10th percentile acc) | |
| | In-distribution | Out-of-distribution | In-distribution | Out-of-distribution |
| ERM | 89.8 (0.8) | 66.6 (1.6) | **72.0** (0.1) | 54.2 (0.8) |
| CORAL | - | - | 71.7 (0.1) | 53.3 (0.0) |
| DANN | - | - | 71.7 (0.1) | 53.3 (0.0) |
| Pseudo-Label | **90.3** (0.5) | 66.9 (2.6) | 71.6 (0.1) | 52.3 (1.1) |
| Masked LM | 89.4 (1.2) | 65.7 (2.3) | 71.9 (0.4) | 53.9 (0.7) |
| ERM (fully-labelled) | 89.9 (0.1) | 69.4 (0.6) | 73.6 (0.1) | 56.4 (0.8) |
| **ICON (Ours)** | 89.7 (0.1) | **68.8** (1.3) | 71.9 (0.1) | **54.7** (0.0) |

Table A2: Supplementary to Table 1. The in-distribution (ID) and out-of-distribution (OOD) performance of each method on each applicable dataset. The standard deviations of each method are reported in the brackets. We bold the highest non-fully-labeled OOD results.

# E    Additional Results

**WILDS 2.0**. In Table A2, we included the standard deviation on WILDS 2.0 benchmark, as well as additional results that are not included in the main paper due to space constraint. In particular, on datasets with data augmentations, we included results of empirical risk minimization (*i.e.*, train in $S$ with cross-entropy loss) with/without data augmentations. We also included the results of an oracle (ERM fully-labeled) on IWILDCAM and FMOW (-wilds suffix excluded for convenience as in the main paper). On POVERTYMAP, we were not able to reproduce the results of ERM (with or without data augmentations) in the original paper, so we included the results that we obtained using the same code, command and environment.