# OpenReview forum: "Make the U in UDA Matter: Invariant Consistency Learning for Unsupervised Domain Adaptation"
_NeurIPS.cc/2023/Conference — NeurIPS 2023 poster_

### Official Review · Reviewer_CYwX · 2023-06-27

**Soundness:** 3 good
**Presentation:** 3 good
**Contribution:** 3 good
**Rating:** 7
**Confidence:** 3

**Summary:**

This paper presents a novel unsupervised domain adaptation method called invariant CONsistency learning (ICON). ICON is very simple; assuming that labeled source samples and clustered target samples are available, ICON uses BCE losses to make the inner product of the features (softmax-normalized) of any two samples in a mini-batch in each domain close to 1 if they are of the same class/cluster, and 0 otherwise. Evaluation experiments on 10 different benchmark datasets (Office-Home, VisDA-2017, WILDS 2.0) show that ICON achieves superior performance to several representative methods on most of them.

**Strengths:**

The proposed method, ICON, is surprisingly simple and easy to use.

A theoretical property of ICON is discussed; ICON gives the optimal classifier under certain assumptions. (this is a fairly high-level characteristic, and the realism of the assumptions are somewhat questionable, though.)

Various experiments are conducted. Despite its simplicity, ICON achieves sufficiently high accuracy on many datasets.

The paper is generally well-written and easy to follow.


**Weaknesses:**

1. The performance of ICON on Office-Home and VisDA-2017 is inferior to that of SoTA. For example, CDTrans [Xu+, ICLR2022] achieves 88.4% on VisDA-2017 and 80.5% on OfficeHome, both higher than ICON.


2. Since the assumptions underlying the theorem appear to be quite strong, it is questionable to what extent they are valid in practice. (this is discussed more or less in the limitation part in the supplementary material, though.)


3. Intuitively, the principle of ICON (i.e., bringing features within the same class/cluster closer together) seems highly similar to that of contrastive learning, which is also a major approach to unsupervised domain adaptation (e.g. [Shen+, ICML2022 ], [Wang+, TMM2023]). Discussing the differences will highlight the property and uniqueness of ICON.


4. While this may be outside the scope of this paper, it would be interesting to discuss the possibility of extending to more advanced domain adaptation problems, such as universal domain adaptation and source-free domain adaptation. Since ICON (perhaps implicitly) assumes that the number of classes in the source and target are the same, and the impact of the number of clusters on accuracy is significant (see Table 2). So the performance of ICON on universal domain adaptation, where the number of classes cannot be assumed to be the same, may not be as promising. Application to source-free domain adaptation is also non-trivial.


[Xu+, ICLR2022] CDTrans: Cross-domain Transformer for Unsupervised Domain Adaptation

[Shen+, ICML2022] Connect, Not Collapse: Explaining Contrastive Learning for Unsupervised Domain Adaptation

[Wang+, TMM2023] Cross-Domain Contrastive Learning for Unsupervised Domain Adaptation


**Questions:**

I have no specific questions that I would like the authors to answer, but it would be great if I could read some discussions on the weaknesses I mentioned above, especially on 1. the comparisons with SoTA.

**Limitations:**

I found a discussion of that limitation in the supplemental material, and it is very reasonable.

---

> ### Author Rebuttal · Authors · 2023-08-09
>
> Thank you for the in-depth review. We address all weaknesses below.
>
> **W1 - SoTA > ICON?** ICON is SoTA on the ResNet-50 backbone. UDA performance is sensitive to backbone choice. We choose the most classic and widely used ResNet-50 to demonstrate the superiority of ICON. In contrast, CDTrans uses the much larger DeiT-base (more than tripling the number of parameters of ResNet-50). Hence the results are not comparable. We also highlight that ICON is agnostic to the backbone choice, and we will try transformer-based backbones in future work.
>
> **W2 - Strong assumption.** The assumptions are actually fundamental for UDA.
> - Assumption 1 corresponds to the classic clustering assumption (or low-density assumption), a necessary assumption for learning with unlabeled data [3]. In fact, UDA methods (and semi-supervised learning in general) commonly leverage feature pre-training and data augmentations to help fulfill the assumptions.
> - Assumption 2 is necessary to rule out corner cases, *e.g.*, in Figure 4b, without additional prior, there is no way for any UDA method to tell if each one of the blue cluster (unlabeled) should be pseudo-labeled as "0" or "1". In practice, these corner cases can be avoided by collecting diverse unlabeled data (lines 213-215), which is generally abundant, *e.g.*, collecting unlabeled street images from a camera-equipped vehicle cruise.
>
> **W3 - Differences with methods based on contrastive learning.** Yes, our method can be viewed as contrastive learning. The differences with previous methods lie in ***what to contrast***. For example, [Shen+, ICML2022] contrasts augmented samples, *i.e.*, a sample under different augmented views shares similar features, and different samples have dissimilar features. [Wang+, TMM2023] contrasts ***cross-domain*** sample pairs (one from source domain $S$ and the other from target domain $T$), *i.e.*, pairs from the same class share similar features (and vice versa). Unfortunately, they still generate $T$ pseudo-labels based on $S$ supervision like self-training methods, and hence are prone to spurious correlations (lines 52-57). Our ICON contrasts ***in-domain*** sample pairs (both samples from $S$ or $T$), *i.e.*, pairs from the same class in $S$ or cluster in $T$ share similar features (and vice versa). In this way, our cluster labels in $T$ only capture the inherent distribution of $T$, which helps remove spurious correlations (lines 66-80).
>
> **W4 - Extension to more advanced DA problems**. Interesting question.
> 1. In Universal DA, there exist classes that only appear in $S$ and $T$. Still, all classes in $S$ and $T$ are typically under a common task, *e.g.*, [1] uses animal classification in its motivating example. Hence this setting meets the condition of ICON (lines 27-32): any sample is generated from a causal feature determining the class identity (*e.g.*, animal appearance) and an environmental feature (*e.g.*, background). From a technical perspective, the extension is not difficult: instead of setting the cluster number in $T$ as the class number in $S$, one can estimate the cluster number (*e.g.*, with semi-supervised k-means in [2]), or use density-based clustering (*e.g.*, DBSCAN) that does not require cluster number.
> 2. Source-Free DA (SFDA) loses access to the data in $S$, *i.e.*, the goal is to adapt a model trained in $S$ with unlabeled data in $T$. Part of ICON can be extended to this setting: In Eq. 2, one can use any loss function in SFDA as $\mathcal{L}_{st}$ and add our consistency loss $\mathrm{BCE}(T,f)$ to it. However, the invariance constraint is no longer applicable without data in $S$.
>
> We will try these settings for future work.
>
> [1] Kaichao You, et al. Universal domain adaptation. CVPR 2019.
>
> [2] Kai Han, et al. Learning to discover novel visual categories via deep transfer clustering. ICCV 2019.
>
> [3] Van Engelen JE, Hoos HH. A survey on semi-supervised learning. Machine learning. 2020 Feb.

---

> > ### Comment · Reviewer_CYwX · 2023-08-16
> >
> > Thanks to the authors for their replies. Regarding #1, I agree with the authors' comments but at the same time would be interested to see if ICON shows a similar advantage with the transformer backbone, as I am not sure if it does.

---

> > > ### Author Response · Authors · 2023-08-17
> > >
> > > Regarding robustness of ICON against backbone choice, we have additional evidence. While we use ResNet-50 on Office-Home and VisDA-2017 following their standard practice, we have a variety of backbone choices on WILDS 2.0 benchmark (following the standard practice of the leaderboard), which is discussed in Appendix Section E.
> > > - We used the followings pre-trained on ImageNet: ResNet-50 [11] on IWILDCAM, DenseNet-121 [12] on FMOW and Faster-RCNN [9] on GLOBALWHEAT.
> > > - We used the *transformer-based* DistilBERT with pre-trained weights from the Transformers library on CIVILCOMMENTS and AMAZON.
> > > - On CAMELYON17, we used DenseNet-121 [12] pre-trained by the self-supervised SwAV [1].
> > > - On POVERTYMAP and OGBMOLPCBA with no pre-training available, we used multi-spectral ResNet-18 [11] and graph isomorphism network [32] trained with the labelled samples in the source domain S, respectively.
> > >
> > > In Table 1, we achieve SOTA on all datasets with different backbones, which is a strong proof that ICON's effectiveness is agnostic to backbone.

---

### Official Review · Reviewer_gs8L · 2023-07-03

**Soundness:** 3 good
**Presentation:** 2 fair
**Contribution:** 3 good
**Rating:** 5
**Confidence:** 4

**Summary:**

This paper proposes ICON (Invariant CONsistency learning), a method to utilize the distribution of unlabeled target data in the UDA task.
And it obtains stable performance improvements over 8 UDA tasks.

**Strengths:**

1. The idea of the article is simple, but it makes sense that the distribution of unlabeled target data does have an impact on UDA tasks.
2. In terms of learning the distribution of unlabeled data, this paper has some novelty in using pseudo labels from low-dimension clustering.
3. This article provides sufficient experiments, and the effectiveness of ICON has been verified under various UDA settings

**Weaknesses:**

Essentially, I think that the core of the ICON method is a pseudo label-based contrastive learning technology, where the pseudo label is from low-dimension clustering. And the authors provide some experiments about the rationality of the ICON, like Fig. 5,6,8. But none of this fundamentally or theoretically explains why low dimensional pseudo labels are more accurate.

**Questions:**

1. Could you provide more analysis about "why low dimensional pseudo labels are more accurate?"
2. If the low-dimensional pseudo labeling works better, why do you still use high dimensions in the final validation?

**Limitations:**

The author has provided limitations in their appendix.

---

> ### Author Rebuttal · Authors · 2023-08-09
>
> Thanks for the constructive feedback. We address all questions below.
>
> **Q1 - Why low-dimensional pseudo-labels are more accurate?**
> We first clarify that our cluster labels are not conventional pseudo-labels, because they are not aligned with the classes in the source domain (details in Reviewer Q3G5-Q4).
> We don't intend to claim that our low-dimensional cluster labels are more accurate (than conventional pseudo-labels). Instead, cluster labels in the target domain provide *additional* supervision *on top of* pseudo-labels to address their limitation, *e.g.*, we empirically show that cluster labels correct pseudo-labeling error in Figure 6b. We also have illustrative and theoretical explanations:
> - In lines 52-57, we illustrate the limitation of conventional pseudo-labels due to spurious correlation in the source domain.
> - In lines 58-65, we explain how cluster labels provide *additional* supervision, and we show how they remove the spurious correlation in lines 66-80.
> - In Section 4, we present a theoretical analysis.
> - We also highlight that "low-dimensional'' is cherry-on-top instead of bread-and-butter for ICON. Please refer to Reviewer cknk-W1 for results and explanation.
>
>
> **Q2 - Why use high dimensions in final validation?**
> This is for practicality and fair comparison: The best-performing dimensionality reduction methods (*e.g.*, t-SNE or UMAP) need to process all data at once. If we train a classifier with low-dimensional features, the final validation must be performed in an offline manner, *i.e.*, having access to all test data to compute low-dimensional features. This is extremely restrictive and impractical, as such a model cannot be deployed to predict online/streaming data. It is also unfair to compare the performance of it against previous methods, which do not have the restriction. Hence we only use dimensionality reduction for clustering the unlabeled training data.

---

### Official Review · Reviewer_RV7L · 2023-07-04

**Soundness:** 2 fair
**Presentation:** 3 good
**Contribution:** 2 fair
**Rating:** 5
**Confidence:** 4

**Summary:**

This paper deals with unsupervised domain adaptation problem. This paper focuses on how to exploit the inherent distribution of target domain to improve the adaptation performance. In detail, it trains two classifier: one is on source domain and the other one is on target domain. Each classifier is trained with BCE loss on a pair of source samples or target samples. For source domain, the same ground-truth label of a pair of samples indicates a positive pair; for target domain, the same cluster label (obtained through clustering) of a pair of samples indicates a positive pair. Finally, the intersection of two classifier is exploited to form the optimal classifier. Additional cross-entropy loss and self-training loss are added to the objective to supervise the adaptation process. Experiment results on various benchmarks validate the effectiveness of proposed method.

**Strengths:**

- The idea is interesting and reasonable to an extent.
- The paper is generally easy to understand and follow.
- The method has been tested on various UDA benchmarks.

**Weaknesses:**

- In one of the claimed contributions, the authors state that their method ICON is orthogonal to previous UDA methods. But I don't see any evidence showing that previous UDA methods combined with ICON should achieve a better adaptation performance.

- In my opinion, it is not suitable to claim outperforming all the conventional methods on WILDS 2.0 UDA benchmark without fully evaluating existing previous works.

- It is interesting to use REx to realize the goal of the second line of Eq. (2). I just wonder what if we simply share the $f$
 between two domains. Will it achieve similar performance? I would like to see such ablations.

- For the experiments, the authors should compare with more recent SOTA UDA methods. The current status of comparisons listed in Table 1 is unsatisfactory and may not justify the performance superiority of proposed method.

- As shown in Fig. 5, the model seems to be sensitive to the hyper-parameter selection. As we know, it is hard to perform hyper-parameter selection in UDA as we don't have labels of target domain data. I feel concerned about whether this work can be really useful in practice.

- The details about how to utilize the self-training loss are not clear. The ablations for self-training terms are missing. Without providing such ablations, we cannot know if the improvement really comes from what the authors claimed.

**Questions:**

See the weakness part.

---

> ### Author Rebuttal · Authors · 2023-08-09
>
> Thanks for the in-depth review. We will address all weaknesses.
>
> **W1 - Orthogonality of ICON.**
> Sorry for the misleading term. We intend to mean that our ICON loss can be plugged into different self-training baselines ($\mathcal{L}_{st}$ in Eq. 2).
> We discussed the choice of the self-training baselines in Appendix Section E under self-training details. For example, on Office-Home and VisDA-2017, we use FixMatch, and greatly improve the accuracy from 69.1% and 76.6% (FixMatch) to 75.4% and 87.4% (FixMatch + ICON) on the two datasets, respectively. We will clarify this in the revision.
>
> **W2 - Claim on WILDS 2.0.**
> Thanks for the suggestion. We will revise accordingly.
>
> **W3 - Ablation on sharing $f$.**
> We have tried sharing $f$ without REx, and it’s worse in Table 3 (ICON-INV).
>
> **W4 - Table 1 may not justify ICON.** We gracefully disagree.
> - For Office-Home and VisDA-2017, we've already included the best-performing UDA methods on ResNet-50 in Table 1. We supplement it with a more comprehensive list and results on ResNet-101 in Appendix Table 3 and 4. Notably, ICON even outperforms the previous SOTA on the much deeper ResNet-101.
> - For WILDS 2.0 benchmark, we included all previous results in its official leaderboard. We also tried implementing recent SOTAs on WILDS2.0. However, without official implementation, it is indeed a nontrivial task to fairly reproduce them as what they claimed to be: they are outperformed by ERM, or even not applicable (*e.g.*, on text/graph data). Hence we feel that it may be disrespectful to include results by us, *e.g.*, the result of recent CST on applicable datasets are shown below. This is in line with WILDS 2.0 organizers' observation that ERM as a naive baseline usually has the best performance, which is exactly the point of WILDS 2.0 as a diagnosis of the ever-overlooked issues in UDA. Hence we believe that dominating this challenging leaderboard strongly justifies the superiority of our ICON.
>
> | Method | iWildCAM | Camelyon | FMoW |
> | ------ | -------- | -------- | ---- |
> | ERM    | 32.2     | 82.0     | 34.8 |
> | CST    | 31.3     | 74.5     | 32.6 |
>
> **W5 - Hyper-parameter ablations.**
> Our work is as practical as other UDA methods: We apologize that our presentation on invariant weight $\beta$ may mislead you. It is standard in the IRM community to choose a small $\beta$ (*e.g.*, 0.1). Then, tuning $\alpha$ follows the same process as other self-training UDA methods.
>
> **W6 - Missing details and ablation for self-training loss.**
> Sorry for the confusion. We include self-training loss details in Appendix Section B.3 and E. The ablation for self-training loss weight $\alpha$ is in Figure 5. Please also refer to Reviewer cknk-W1 for a discussion on ICON's improvement over self-training.

---

### Official Review · Reviewer_Q3G5 · 2023-07-05

**Soundness:** 4 excellent
**Presentation:** 3 good
**Contribution:** 3 good
**Rating:** 7
**Confidence:** 3

**Summary:**

This paper proposes a new UDA method which strives to produce a consistent classifier for labels in source domain and clusters in target domain. Specifically, this paper introduces an auxiliary task for distinguishing whether the input image pair share the same class/cluster or not. This binary classification task would encourage the features from the same class to be similar and features from different classes to be dissimilar. The groundtruth for this binary classification task on target pair is determined by performing clustering on target features. By combining this binary classification task with supervised learning on source domain and self training on target domain, this method obtains state-of-the-art performance on multiple challenging UDA benchmark.

**Strengths:**

1. This paper utilizes clustering on target features to determine whether the target image pair share the same class. Compared to previous self-training methods, this design better reduces the noise in the pseudo labels and alleviate overfitting on incorrect pseudo labels.
2. The proposed method obtains state-of-the-art performance on many challenging UDA tasks. A sufficient ablation study and analysis clearly depicts the advantage of the proposed method, which would provide great insight for furtehr works.

**Weaknesses:**

Basically I believe the method proposed method greatly meets its motivation and sufficient analysis has proven its effectiveness. The introduction part could be further polished to emphasize the core idea better. For figure 1, the lack of comparison on t-SNE of baselines might make the reader difficult to understand the advantage of ICON.

**Questions:**

1. In line41 there are double commas.
2. The BCE loss is only applied on source-source pair or target-target pair. What if applying it on source-target pair? Does it bring better performance?
3. How the pseudo labels for $L_{st}$ is obtained? A short introduction should be included in the implementation details.
4. Now that the main idea is to utilize the cluster label to group target domain features, is it necessary to add a BCE task for UDA classification? Is it possible to utilize the cluster label as the pseudo label for self-training?

**Limitations:**

This paper adequately discussed the limitation of the proposed method and provide an analysis about the possible reasons. No potential negative societal impact is discussed.

---

> ### Author Rebuttal · Authors · 2023-08-09
>
> Thanks for the constructive feedback. We will fix the typos in Q1 and address all concerns below.
>
> **W - Lack of baselines t-SNE in Figure 1.** Sorry for the confusion. Actually, the goal of Figure 1 is not to compare our ICON with baselines, but to depict the condition where a model generalizes. In Figure 2, we further discuss how ICON achieves this condition and hence improves existing baselines.
>
> **Q2 - BCE between source-target pair?** Thanks for the suggestion. We tried and observed comparable performance (75.3% on Office-Home, 87.0% on VisDA-2017). Specifically, for a source-target pair, we compared the ground-truth label (source sample) and pseudo-label (target sample) to get the binary label for BCE loss (*i.e.*, $b$ in Eq. 1). This is because we cannot directly compare the ground-truth label and cluster label (see Q4). We postulate that the self-training loss ($\mathcal{L}_{st}$ in Eq. 2) achieves similar effect with the source-target BCE loss, *i.e.*, bringing each target sample closer to its pseudo-label class center.
>
> **Q3 - How to obtain $\mathcal{L}_{st}$.** Sorry for the confusion. We discussed this in Appendix Section B.3 and E (under self-training details). We mainly used FixMatch and Noisy Student loss as $\mathcal{L}_{st}$, where pseudo-labels are generated on weakly-augmented target domain samples, and the model is trained to predict the pseudo-labels given strongly-augmented samples. We will give a short introduction in the main text in revision.
>
> **Q4 - Cluster label as pseudo label?.**
> Unfortunately, this is not possible, because the cluster labels (*e.g.*, cluster 1 or 2) are not aligned with the class labels (*e.g.*, "bed'' or "clock'') as discussed in lines 155-156, and explicit alignment (*e.g.*, assigning cluster 1 to "bed'') reduces our method to standard self-training. Hence it is necessary to use the BCE loss to train the classifier, such that its decision boundary simultaneously separates the classes in the source domain and the clusters in the target domain (Figure 2d).

---

> > ### Comment · Reviewer_Q3G5 · 2023-08-20
> >
> > The rebuttal has addressed all my concerns. After viewing the rebuttal and other reviewers' comments, I believe this paper worth acceptance for top-level conferences like Neurips. Thus, I will keep my rating to accept.

---

### Official Review · Reviewer_cknk · 2023-07-05

**Soundness:** 3 good
**Presentation:** 4 excellent
**Contribution:** 3 good
**Rating:** 6
**Confidence:** 3

**Summary:**

The authors propose an unsupervised domain adaptation method, ICON. The algorithm is similar to self-training on the unlabeled target data, but at the start of each epoch, the unlabeled data are first projected from feature space to a reduced-dimension space and clustered. An auxiliary loss enforces consistent pseudo-labels within clusters. The method attained state-of-the-art on the WILDS unlabeled benchmark [1].

[1] Sagawa, S., Koh, P. W., Lee, T., Gao, I., Xie, S. M., Shen, K., ... & Liang, P. (2021). Extending the WILDS benchmark for unsupervised adaptation. *arXiv preprint arXiv:2112.05090*.

**Strengths:**

- ICON’s strong empirical performance is striking, especially on WILDS; the authors outperform similar self-training-style methods, e.g.. vanilla self-training, FixMatch, and Noisy Student. Notably, ICON attains this performance without using aggressive data augmentations.
- The paper evaluates on ten datasets across 3 modalities, and evaluations are conducted over multiple seeds.
- The presentation is good, and the experiments are notably well-documented.

**Weaknesses:**

- Given its similarity to other self-training methods, it seems important to analyze, whether through experimental ablations or theory, why ICON outperforms vanilla self-training so strikingly. One unique aspect of ICON is that the unlabeled consistency loss uses clusters computed in a reduced dimension space; this seems important in ablations (Table 2). The authors motivate this by stating that dimensionality reduction acts to suppress environmental features and highlight the causal feature (L138), but this statement seems unsupported to me. Could the authors clarify theoretically why this is?
- The theory in Section 4 makes two strong assumptions, to the effect of (1) the classes in T are cleanly separated by clusters, and (2) if the model separates classes correctly in S and clusters in T, then the classes in T are predicted correctly, as prediction uses the invariant feature.

**Questions:**

- The authors present ICON as orthogonal to existing UDA approaches, e.g. consistency regularization using views from data augmentations. Could the authors share how ICON performs when combined with strong data augmentations, e.g. RandAugment?
- On page 4, the authors state that any clustering algorithm can be used, so long as the number of clusters should be equal to the number of classes. This is a useful ablation; in general, understanding which representation space and how to cluster seems important. Did the authors have k-means experiments to support this?
- I could not find the UMAP hyperparameters in the appendix as the main paper suggested; in particular, what was the output dimension used?
- The results of iWildCam2020-WILDS and CivilComments-WILDS are a bit unaligned with the theory, as in these cases, the unlabeled data is *not* drawn from the test distribution. It would be great if the main paper could comment briefly on this discrepancy.

**Limitations:**

The authors note that the assumptions made in Section 4 are restrictive.

---

> ### Author Rebuttal · Authors · 2023-08-09
>
> Thanks for the in-depth comments and suggestions. We will address all concerns below.
>
> **W1 - Why ICON outperforms self-training.** We clarify that the key to ICON's success is invariant consistency instead of dimensionality reduction with UMAP.
> - For empirical evidence, we perform additional experiments and reformat the results in Table 2 to form Table r1 below. Note that we used FixMatch as a self-training baseline on Office-Home and VisDA-2017. It shows that consistency loss brings the most improvement, followed by invariance constraint, and lastly UMAP.
> - For theoretical analysis, we analyze the failure of self-training in Figure 2a (lines 52-57) and discuss how clusters in the target domain provide additional supervision to prevent the failure (lines 58-65). Then we propose consistency loss (lines 66-71) and invariance constraint (lines 72-80) to incorporate this supervision. We clarify that UMAP is simply a pre-processing technique to improve clustering, which is shown to benefit classification tasks (*i.e.*, highlights causal feature) supported by rigorous theoretical results [1].
>
> Table r1: Ablations on each ICON component. Supplement to Table 2.
> | Method                                            | Office-Home | VisDA-2017 |
> | ------------------------------------------------- | ----------- | ---------- |
> | FixMatch                                          | 69.1        | 76.6       |
> | FixMatch + Consistency                            | 74.1        | 82.0       |
> | FixMatch + Consistency + Invariance               | 75.2        | 86.5       |
> | FixMatch + Consistency + Invariance + UMAP (ICON) | 75.4        | 87.4       |
>
>
> **W2 - Strong assumptions.** The assumptions are actually fundamental for UDA.
> - Assumption 1 corresponds to the classic clustering assumption (or low-density assumption), a necessary assumption for learning with unlabeled data [2]. In fact, UDA methods (and semi-supervised learning in general) commonly leverage feature pre-training and data augmentations to help fulfill the assumptions.
> - Assumption 2 is necessary to rule out corner cases, *e.g.*, in Figure 4b, without additional prior, there is no way for *any* UDA method to tell if each one of the blue cluster (unlabeled) should be pseudo-labeled as "0" or "1". In practice, these corner cases can be avoided by collecting diverse unlabeled data (lines 213-215), which is generally abundant, *e.g.*, collecting unlabeled street images from a camera-equipped vehicle cruise.
>
> **Q1 - ICON + strong data augmentations.** Actually, self-training baselines already leverage strong data augmentations. For example, FixMatch in Table r1 enforces the predictions on strongly augmented samples (using RandAugment) to be similar to those on weakly augmented samples.
> We also highlight that ICON's success does not rely on strong augmentations, *e.g.*, ICON still outperforms on text modality (CivilComments and Amazon), where strong augmentations are not available.
>
> **Q2 - k-means experiment.** We tried k-means and got 85.6% on VisDA-2017. We postulate that rank-statistics is better because its online clustering provides up-to-date cluster assignments. In the future, we will try other more recent online clustering methods, *e.g.*, an optimal transport-based method [3].
>
> **Q3 - UMAP hyperparameters.** Sorry for omitting it. We used 50 as the output dimension and used the default values in the official UMAP repo for all other hyperparameters.
>
> **Q4 - Datasets unaligned with theory.** Sorry for the confusion. The two datasets are indeed aligned with the theory. Note that ICON works by learning the causal feature $\mathbf{c}$ and discarding the environmental feature $\mathbf{e}$ (lines 66-80). Hence the theory holds as long as the definition of $\mathbf{c}$ and $\mathbf{e}$ is consistent across the training and test data, *e.g.*, when the task is digit classification as in Figure 2, $\mathbf{c},\mathbf{e}$ corresponds to digit shape and digit color, respectively. In iWildCam-WILDS and CivilComments-WILDS, while the unlabeled data is not drawn from the test distribution, the task is the same across labeled, unlabeled, and test data (*e.g.*, animal classification), *i.e.*, the definition of $\mathbf{c},\mathbf{e}$ remains consistent. We will revise the main paper to clarify this.
>
> [1] Leland McInnes, et al. UMAP: Uniform Manifold Approximation and Projection for Dimension Reduction.
>
> [2] Van Engelen JE, Hoos HH. A survey on semi-supervised learning. Machine learning. 2020 Feb.
>
> [3] Mathilde Caron, et al. Unsupervised learning of visual features by contrasting cluster assignments. NeurIPS 2020.

---

> > ### Comment · Reviewer_cknk · 2023-08-16
> >
> > Thanks to the authors for their response.
> >
> > - **W1-W2.** Thanks to the authors for their response, particularly the empirical results showing that UMAP is helpful (rows 3 v. 4 in Table r1). However, I feel that my question has not been adequately addressed. My question was about the claim made that dimensionality reduction "highlights" causal features and "suppresses" environmental features (L138, LL207-208). If I understand correctly, this claim is not studied in the theory, which assumes that UMAP has already been applied to satisfy Assumption 1. I've also looked through the reference [1] that the authors gave; I do not see a justification for the authors' claim in this work. Perhaps the authors could point me to a particular page/line of [1] or Section 4 / Appendix D if I've misunderstood. It's important that this claim is justified, *particularly* since Table r1 and the authors' experiment with k-means suggests that the dimensionality reduction / clustering steps do contribute to the empirical success of ICON.
> >
> > - **Q1.** I agree with what the authors have responded -- it is indeed impressive that ICON does not require strong data augmentations --  but I'd like to clarify my question. I was asking for empirical evidence validating L82, which states that ICON is orthogonal to data augmentation techniques and can be combined with them (presumably, with additive gains). Do the authors have experiments verifying this?
> >
> > - **Q2-4.** Thanks for the clarifications; these have answered my questions.

---

> > > ### Author Response · Authors · 2023-08-17
> > >
> > > Thanks for clarifying the questions. We will address them below.
> > >
> > > **W1-W2.** Yes, as you point out, we do not aim to study/prove this claim in our theory. We use dimensionality reduction as a practical method to approach the conditions in Assumption 1 (*i.e.*, clustering unlabeld samples), which can be justified in the following ways:
> > > 1) In Section 5.1-5.2 of [1], UMAP is empirically validated to faithfully capture the local data structure (e.g., high k-NN classification accuracy) as well as the global structure (e.g., more meaningful global relationships between different classes) in a low-dimensional space. Such structural information is however elusive in the original high-dimensional space due to the curse of dimensionality. Hence we say that UMAP highlights the causal feature, *i.e.*, clustering same-class samples (local structure) and pushing away different-class samples (global structure).
> > > 2) Dimensionality reduction is already extensively used in classification tasks. For example, feature bottlenecking, as a learnable dimensionality reduction method, is adopted in UDA methods such as CST [29], and CLIP-Adapter for fine-tuning vision language models on downstream tasks. Another example is t-SNE, a standard visualization method in classification tasks to evaluate feature quality (*i.e.*, whether same-class samples are clustered).
> > >
> > > **Q1.** We clarify that L82 means that ICON can be combined with different self-training methods. We include the self-training details in Appendix Section E. In Table 1, we improve all self-training methods. For data augmentations, we follow the standard practice in all benchmarks to facilitate fair comparison with previous works.

---

> > > > ### Comment · Reviewer_cknk · 2023-08-19
> > > >
> > > > Thanks for the clarifications! It would be appreciated if this discussion was reflected in the manuscript, so that the claims are more precise. In particular, the claims about causal features should be made carefully.
> > > >
> > > > My questions have been addressed.

---

> ### Comment · Area_Chair_rSfh · 2023-08-16
> **Feedback to the authors**
>
> Dear R#cknk,
>
> The authors have provided a rebuttal. Can you please confirm whether it addresses the questions you had asked. We have a brief period for interaction between reviewers and authors.
>
> Best,
> AC

---

### Decision · Program_Chairs · 2023-09-21

**Decision:**

Accept (poster)

**Comment:**

The final views on the paper ranged from borderline accept to accept.

The main concerns of one of the reviewers  (borderline accept) were that the claims for causality needs to be carefully made. The point about the same class features being closer and the different class features being further does not imply causality and as raised by the reviewer, this needs to be clarified. The point by the other reviewer with borderline views are also valid, in that the claims of orthogonality are not generally proven. The specific empirical method can be provided as indications of complementarity (rather than orthogonality).

On balance, it is felt that the paper does provide thorough empirical analysis and the reviewers are convinced by the rebuttal. It is therefore recommended that the paper be accepted. It is strongly recommended to revise the paper by taking all the points raised by reviewers into account while preparing the final version of the paper.